# FiSeR: Fine-Grained Source Representations for Cross-Domain AI Image Detection

Shan Zhang [* 1 2 4]   Yongxin He [* 3 4]   Mingming Zhang [1]   Huiwen Tian [3 4]   Lei Ma [1 2 4]

## Abstract

Real-world synthetic image detectors often generalize poorly under domain shift despite strong in-domain performance. Using unsupervised UMAP projections, we find that natural and synthetic features remain partially separable on unseen datasets, yet performance still drops, suggesting that the classification head overfits to training-domain artifacts. Therefore, the key is to learn more transferable representations so that the decision criterion is more stable and robust to domain shifts. Based on the structural fact that synthetic images are produced by diverse generators, we propose a hierarchical contrastive learning framework that improves the separability between natural and synthetic images while preserving generator identity information. It jointly optimizes (i) a coarse contrastive objective between natural and synthetic images and (ii) a fine contrastive objective among synthetic images using generator identities. Trained on Wild-Fake, our method achieves an average AUROC gain of +10.22 on cross-domain evaluation over Chameleon, AIGIBench, Community Forensics, and GenImage under the same settings as the strong baseline DIRE. For few-shot adaptation, we freeze the backbone and fit an SVM head on 10 labeled samples per class, improving AUROC by +10.64 on AIGIBench and +17.41 on Chameleon, averaged over 12 widely used detectors. Our code is publicly available at: https://github.com/heyongxin233/FiSeR.

---

[*]Equal contribution  [1]The Key Laboratory of Cognition and Decision Intelligence for Complex Systems, Institute of Automation, Chinese Academy of Sciences, Beijing, China [2]School of Artificial Intelligence, University of Chinese Academy of Sciences, Beijing, China [3]Key Lab of Intelligent Information Processing, Institute of Computing Technology, Chinese Academy of Sciences, Beijing, China [4]University of Chinese Academy of Sciences, CAS, Beijing, China. Correspondence to: Lei Ma <lei.ma@ia.ac.cn>.

*Proceedings of the 43rd International Conference on Machine Learning*, Seoul, South Korea. PMLR 306, 2026. Copyright 2026 by the author(s).

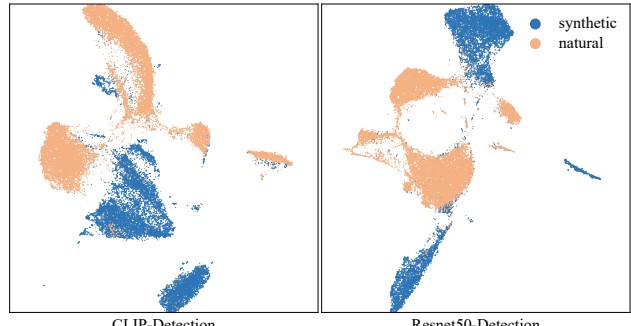

*Figure 1.* Unsupervised UMAP visualization of intermediate representations for CLIP-Detection and ResNet50-Detection, trained on WildFake and evaluated on Chameleon.

*Table 1.* AUROC on Chameleon. Direct: evaluate on Chameleon with the binary head trained on WildFake, without any adaptation. SVM (10-shot): fit an SVM boundary using 10-shot labeled samples from Chameleon; numbers in parentheses denote absolute AUROC gains.

|  | **Direct** | **SVM (10-shot)** |
|---|---|---|
| ResNet50-Detection | 56.04 | **92.38** (+36.34) |
| CLIP-Detection | 60.81 | **85.39** (+24.58) |

## 1. Introduction

Generative models (Ramesh et al., 2021; Betker et al., 2023; Labs et al., 2025) can now produce realistic images at scale. Detecting synthetic images has become important for misinformation control and content moderation. Yet most detectors (Liu et al., 2020; Wang et al., 2020; Yan et al., 2024a) are trained and tested in a closed setting with fixed generators and stable data. In practice, generators change quickly and the image content and sources also shift. Under this domain shift, detection accuracy drops sharply. Cross domain benchmarks such as AIGIBench (Li et al., 2025b) show that this remains true even for strong detectors.

We study why this happens by analyzing intermediate layer features of popular detectors (He et al., 2016; Radford et al., 2021) on new datasets. Using UMAP (McInnes et al., 2018), we find that natural and synthetic images are still partly separable in feature space, as shown in Figure 1. The separable structure does not disappear, but the final performance still drops, as reported in Table 1. This suggests that the classification head is a main failure point. It tends to fit domain

specific artifacts, so its decision boundary does not match the feature geometry after the domain shifts.

Based on this observation, we aim to learn features whose geometry is more stable across domains. Given the structural fact that synthetic images are produced by different generators, preserving generator-source structure in the representation space can reduce reliance on domain-specific spurious cues and improve transferability. We therefore propose a hierarchical contrastive learning framework that improves the separability between natural and synthetic images while preserving generator identity information. The framework jointly optimizes two objectives: (i) a coarse contrastive objective between natural and synthetic images, and (ii) a fine contrastive objective among synthetic images grouped by generator identity. We also introduce a $k$-nearest neighbors graph homophily score to quantify representation quality by measuring how often nearest neighbor edges connect samples from the same source.

At inference time, we freeze the backbone and attach a lightweight classification head on top of the extracted features, including $k$-NN, a support vector machine (SVM), or a linear classifier. We train on WildFake and test on Chameleon, AIGIBench, Community Forensics, and GenImage without using any images from these target datasets during training. Under the same protocol as DIRE (Wang et al., 2023a), we improve the average AUROC by 10.22 points. With only 10 labeled samples per class from the target domain, fitting a lightweight head improves AUROC by 10.64 on AIGIBench and 17.41 on Chameleon, averaged across 12 widely used detectors.

In summary, our contributions are threefold: (I) revealing a key observation in cross-domain generative image detection: performance drops do not necessarily imply a collapse of intermediate representations, as the feature space can still preserve exploitable separable structure; (II) proposing a hierarchical supervised contrastive learning framework that enforces coarse- and fine-grained source constraints to learn more fine-grained and more generalizable source-related representations; (III) proposing a $k$-NN graph homophily metric to quantify representation separability; building on this, we achieve the best results on multiple benchmarks and substantially improve cross-domain detection of existing detectors with no extra training or only lightweight training.

## 2. Related Work

### 2.1. Benchmarks for AI-Generated Image Detection

Early benchmarks such as AIGCDetect-Benchmark (Zhong et al., 2023) and CNNSpot (Wang et al., 2020) mainly aggregate samples from roughly 16 generative models, including ProGAN (Karras et al., 2017), to support basic cross-model generalization tests. However, their scale and content complexity are limited. To broaden coverage, later benchmarks scale up the data and model diversity, including DE-FAKE (Sha et al., 2023), CiFAKE (Bird & Lotfi, 2024), DiffusionDB (Wang et al., 2023b), and ArtiFact (Rahman et al., 2023). GenImage (Zhu et al., 2023) and Fake2M (Lu et al., 2023) further construct million-level paired real and synthetic data with wider object categories. To address insufficient detector generalization, WildFake (Hong & Zhang, 2024) expands category and style diversity and organizes the data hierarchically. Recent work emphasizes evaluation under realistic conditions: AIGIBench (Li et al., 2025b) introduces in-the-wild real and synthetic samples from social media and unknown distributions such as CommunityAI and SocialRF, which substantially increases test difficulty. To probe the upper bound of detectors on highly photorealistic synthesis, Chameleon (Yan et al., 2024a) builds an extreme high-fidelity test set with strong visual deceptiveness. Community Forensics (Park & Owens, 2025) focuses on source traceability and model diversity, collecting outputs from 4,803 open-source or commercial generative models.

### 2.2. Methods for Detecting AI-Generated Images

AI image detection methods have evolved rapidly. Early approaches often rely on explicit spatial artifacts such as color (McCloskey & Albright, 2018), saturation (McCloskey & Albright, 2019). As generators improve, these cues are increasingly suppressed, which limits cross-generator generalization. In the line of spatial texture statistics, methods such as FatFormer (Liu et al., 2024), CO-SPY (Cheng et al., 2025), Secret Lies in Color (Jia et al., 2025), and Deep Image Fingerprint (Sinitsa & Fried, 2024) identify synthetic traces from spatial features. Representative examples include Gram-Net (Liu et al., 2020), which leverages global texture statistics to improve robustness to edits and distribution shifts, and CNNDetection (Wang et al., 2020), which combines strong pre and post processing with data augmentation. To mitigate the erosion of spatial cues, many works such as SPAI (Karageorgiou et al., 2025) turn to frequency-domain and residual artifacts. NPR (Tan et al., 2024b) explicitly encodes adjacent-pixel relations as artifact representations, FreqNet (Tan et al., 2024a) emphasizes sustained attention to high-frequency information, PiD (Fu et al., 2025) uses quantization error as a key feature, and AIDE (Yan et al., 2024a) fuses high-level semantics with low-level artifacts.

In the past two years, a new paradigm has emerged that builds detectors on top of foundation-model features, including LaRE$^2$ (Luo et al., 2024), and FakeInversion (Cazenavette et al., 2024). UniFD (Ojha et al., 2023) trains a general linear classifier on pretrained CLIP ViT features. DIRE (Wang et al., 2023a) constructs features from reconstruction discrepancies of a pretrained ADM and trains a deep classifier. LGrad (Tan et al., 2023) uses gradients

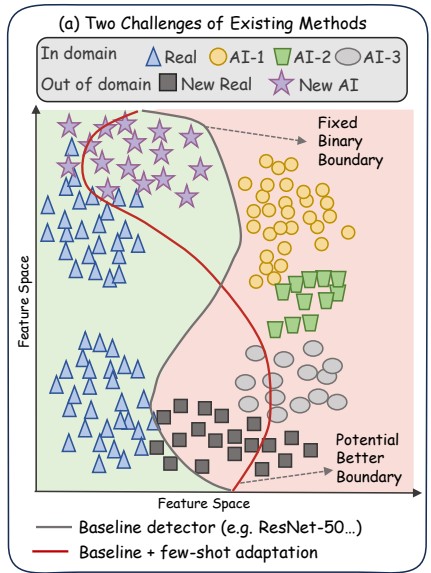
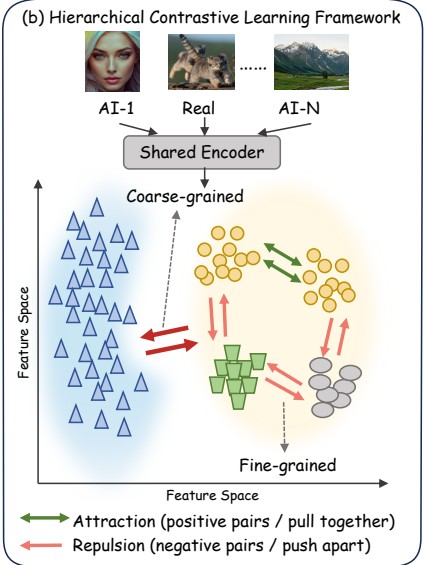
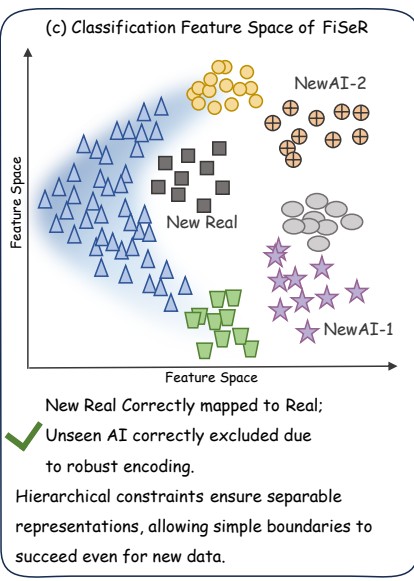

*Figure 2.* **Overview of FiSeR.** (a) Two challenges under distribution shift: (i) in-domain decision boundaries often fail to generalize, misclassifying unseen natural and synthetic samples; and (ii) backbone features can become less separable between natural and synthetic when new generators or new natural sources emerge. (b) FiSeR trains an image encoder with hierarchical supervised contrastive learning, combining a coarse natural–synthetic objective with a fine within-synthetic objective based on generator identity. (c) FiSeR achieves fine-grained separation among seen synthetic sources and improves generalization to unseen synthetic and unseen natural samples.

from a pretrained GAN discriminator as artifact representations. Finally, to handle distribution shifts in real-world settings, methods such as Breaking Semantic Artifacts (Zheng et al., 2024), and ZED (Cozzolino et al., 2024) focus on robust training and rapid adaptation. SAFE (Li et al., 2025a) improves generalization via targeted image transformations. ManiFPT (Song et al., 2024) formalizes artifacts and fingerprints of generative models and analyzes their value for source attribution and recognition. LASTED (Wu et al., 2025a) formulates detection as an identity-style contrastive learning problem. FSD (Wu et al., 2025b) models detection as a few-shot task and achieves fast adaptation to unseen models with very few samples.

## 3. Method

We study the binary classification problem of AI-generated image detection. Given an input image $x$, the goal is to predict a binary label $y(x) \in \mathcal{Y} = \{\text{natural}, \text{synthetic}\}$, where *natural* denotes photographs captured from real-world scenes by cameras or smartphones, and *synthetic* denotes images generated or manipulated by algorithms, including GAN-based, diffusion-based, and autoregressive generation. The motivation, overall architecture, and effectiveness of our method, FiSeR, are shown in Figure 2.

To improve the robustness of synthetic image detection under realistic distribution shifts, we next present three components of our approach: (i) hierarchical supervised contrastive representation learning for learning transferable yet source-discriminative feature representations (Section 3.1), (ii) a

plug-and-play lightweight classification head that adapts the decision boundary under domain shift for inference (Section 3.2), and (iii) $k$-NN graph homophily as a quantitative measure of representation separability (Section 3.3).

### 3.1. Hierarchical Supervised Contrastive Representation Learning

In real-world settings, AI-generated image detection faces persistent distribution non-stationarity. As generators evolve in version and architecture, the test distribution continually shifts, and discriminative cues learned on the source domain can substantially weaken under unseen generators or degradations (Li et al., 2025b). We therefore cast the problem as learning representations that generalize across generators and generation paradigms under ongoing distribution shift and the continual emergence of new generators.

Our design is motivated by an empirical observation: different image formation and generation mechanisms often leave reproducible low-level statistical traces. Natural images, governed by real-world image formation processes, exhibit relatively stable distributions in low-level feature space (Karageorgiou et al., 2025; Zhong et al., 2025), whereas synthetic images may carry model-specific fingerprints (Sinitsa & Fried, 2024; Song et al., 2024). Accordingly, we propose hierarchical supervised contrastive learning to jointly encourage binary separability and structured intra-synthetic representations: (i) a coarse-grained constraint separates natural from synthetic; (ii) a fine-grained constraint promotes intra-source compactness and inter-

source dispersion among synthetic samples, improving the characterization of generation artifacts and enhancing transferability under distribution shifts and unseen generators.

Let $\mathcal{X}$ denote the image distribution. Each sample $x \sim \mathcal{X}$ has a binary label $y(x) \in \mathcal{Y} = \{\text{natural}, \text{synthetic}\}$ and a source label $s(x) \in \{0, 1, \ldots, S\}$, where $s(x) = 0$ indicates natural images without distinguishing camera or device identity, and $s(x) \in \{1, \ldots, S\}$ indexes different generators. We learn an encoder $f_\theta : \mathcal{X} \to \mathbb{R}^d$ and define $\mathrm{sim}(\cdot, \cdot)$ as cosine similarity in the embedding space.

Under the two-level constraints described above, the coarse-grained objective maximizes similarity for pairs sharing the same binary label and minimizes similarity otherwise:

$$
\max_\theta \mathop{\mathbb{E}}_{\substack{x,x' \sim \mathcal{X} \\ y(x)=y(x')}} \left[ \mathrm{sim}(f_\theta(x),\, f_\theta(x')) \right] - \\
\mathop{\mathbb{E}}_{\substack{x,x' \sim \mathcal{X} \\ y(x) \neq y(x')}} \left[ \mathrm{sim}(f_\theta(x),\, f_\theta(x')) \right]. \tag{1}
$$

For the fine-grained objective, we restrict the expectation to pairs satisfying $y(x) = y(x') = \text{synthetic}$, and we encourage similarity for pairs from the same source while minimizing similarity for pairs from different sources:

$$
\max_\theta \mathop{\mathbb{E}}_{\substack{x,x' \sim \mathcal{X} \\ y(x)=y(x')=\text{synthetic}, \\ s(x)=s(x')}} \left[ \mathrm{sim}(f_\theta(x),\, f_\theta(x')) \right] - \\
\mathop{\mathbb{E}}_{\substack{x,x' \sim \mathcal{X} \\ y(x)=y(x')=\text{synthetic}, \\ s(x) \neq s(x')}} \left[ \mathrm{sim}(f_\theta(x),\, f_\theta(x')) \right]. \tag{2}
$$

Equations 1 and 2 specify the desired attraction and repulsion structure in the embedding space. In practice, we optimize these objectives by minimizing corresponding supervised contrastive losses as differentiable surrogates.

Given a mini batch of samples $\mathcal{B} = (x_i)_{i=1}^{B}$ with $B = |\mathcal{B}|$, we denote $y_i = y(x_i)$, $s_i = s(x_i)$, and $\mathbf{z}_i = f_\theta(x_i)$.

For the coarse-grained objective in Equation 1, the positive and negative index sets for $x_i$ are defined as:

$$
\mathcal{P}_\mathrm{y}(i) = \{\, k \in \{1, \ldots, B\} \setminus \{i\} \mid y_k = y_i \,\}, \\
\mathcal{N}_\mathrm{y}^-(i) = \{\, k \in \{1, \ldots, B\} \setminus \{i\} \mid y_k \neq y_i \,\}. \tag{3}
$$

Unlike standard supervised contrastive learning (Khosla et al., 2020), we aggregate all positives into a single positive logit by averaging similarities before exponentiation. This reduces competition among positives in the softmax and improves stability in our experiments. We define the average positive similarity for $x_i$ in the coarse-grained task as:

$$
\bar{\ell}_\mathrm{y}(i) = \frac{1}{|\mathcal{P}_\mathrm{y}(i)|} \sum_{k \in \mathcal{P}_\mathrm{y}(i)} \mathrm{sim}(\mathbf{z}_i, \mathbf{z}_k). \tag{4}
$$

When $|\mathcal{P}_\mathrm{y}(i)| = 0$, we set $\bar{\ell}_\mathrm{y}(i) = 0$. In this case, the positive logit is a constant, and the loss is primarily driven

by negative similarities. The coarse-grained supervised contrastive loss is then defined as:

$$
\mathcal{L}_\mathrm{y}(x_i; \theta) = \\
-\log \frac{\exp(\bar{\ell}_\mathrm{y}(i)/\tau)}{\exp(\bar{\ell}_\mathrm{y}(i)/\tau) + \sum_{k \in \mathcal{N}_\mathrm{y}^-(i)} \exp(\mathrm{sim}(\mathbf{z}_i, \mathbf{z}_k)/\tau)}, \tag{5}
$$

where $\tau > 0$ is the temperature. For the fine-grained objective, we only impose constraints on samples with $y_i = \text{synthetic}$. The corresponding positive and negative index sets are:

$$
\mathcal{P}_\mathrm{src}(i) = \{\, k \in \{1, \ldots, B\} \setminus \{i\} \mid y_k = \text{synthetic}, s_k = s_i \,\}, \\
\mathcal{N}_\mathrm{src}^-(i) = \{\, k \in \{1, \ldots, B\} \setminus \{i\} \mid y_k = \text{synthetic}, s_k \neq s_i \,\}. \tag{6}
$$

The fine-grained loss $\mathcal{L}_\mathrm{src}(x_i; \theta)$ takes the same form as $\mathcal{L}_\mathrm{y}(x_i; \theta)$ by replacing $\mathcal{P}_\mathrm{y}(i)$ and $\mathcal{N}_\mathrm{y}^-(i)$ with $\mathcal{P}_\mathrm{src}(i)$ and $\mathcal{N}_\mathrm{src}^-(i)$. When $|\mathcal{P}_\mathrm{src}(i)| = 0$, we set $\bar{\ell}_\mathrm{src}(i) = 0$. We compute $\mathcal{L}_\mathrm{src}(x_i; \theta)$ only for $y_i = \text{synthetic}$.

Our overall contrastive objective combines the coarse-grained and fine-grained losses, where $\lambda \geq 0$ controls the weight of the fine-grained constraint:

$$
\mathcal{L}(x_i; \theta) = \mathcal{L}_\mathrm{y}(x_i; \theta) + \lambda\, \mathbb{I}[y_i = \text{synthetic}]\, \mathcal{L}_\mathrm{src}(x_i; \theta). \tag{7}
$$

During training, we minimize the batch-averaged loss:

$$
\min_\theta \frac{1}{B} \sum_{i=1}^{B} \mathcal{L}(x_i; \theta). \tag{8}
$$

### 3.2. Lightweight Plug-and-Play Classification Head

After obtaining source-discriminative yet transferable representations, we perform inference-time adaptation by attaching a plug-and-play lightweight head, which can be constructed from a small task-specific labeled support set. Given a backbone encoder $f_\theta$, we construct a feature–label memory:

$$
\mathcal{M} = \{(\mathbf{z}_i, y_i)\}_{i=1}^{M}, \quad M = |\mathcal{M}|, \mathbf{z}_i = f_\theta(x_i), y_i \in \mathcal{Y}, \tag{9}
$$

where $\mathcal{Y}$ denotes the binary label space.

Based on $\mathcal{M}$, we instantiate a discriminative function $g_\phi$, where $\phi$ denotes the instantiated head, either fitted parameters or nonparametric statistics derived from $\mathcal{M}$. Concretely, $g_\phi$ can be a training-free nonparametric rule, such as $k$-NN or prototypical classification, or a lightweight parametric model fit on $\mathcal{M}$ with negligible overhead, such as a linear classifier, or SVM. Given an input image $x$, we compute its embedding $\mathbf{z} = f_\theta(x)$ and obtain class scores:

$$
S(c \mid x) = g_\phi(c; \mathbf{z}), \qquad c \in \mathcal{Y}. \tag{10}
$$

This inference procedure requires no updates to the backbone and uses the frozen encoder together with the support memory $\mathcal{M}$ to instantiate $g_\phi$. We evaluate multiple instantiations of $g_\phi$ in the experiments.

### 3.3. $k$-NN Graph Homophily for Separability

Our experiments show that replacing the parametric classification heads in methods such as ResNet50-Detection and CLIP-Detection with our plug-and-play inference head yields consistent improvements under various distribution shift settings. This suggests that degraded detection performance does not necessarily imply that the representation is ineffective, since the local neighborhood structure in the embedding space may still preserve useful homophily signals. To quantitatively evaluate the separability of different encoders and their intermediate-layer representations, we propose the $k$-NN graph homophily score.

Specifically, given an encoder $f_\theta$ and a set of representations with source labels, we form $\mathcal{Z} = \{(\mathbf{z}_i, s_i)\}_{i=1}^n$, where $\mathbf{z}_i = f_\theta(x_i)$ and $s_i = s(x_i)$. We construct a directed $k$-NN graph in the embedding space. For each sample $x_i$, let $\mathcal{N}_k(i) \subset \{1, \ldots, n\} \setminus \{i\}$ denote the index set of its $k$ nearest neighbors. Here $k$ is the number of neighbors used for homophily evaluation and can be set independently of the $k$ used by the $k$-NN classifier at inference.

We define the proportion of same-source neighbors for $x_i$ as:

$$t_k(i) = \frac{1}{k} \sum_{j \in \mathcal{N}_k(i)} \mathbb{I}[s_j = s_i]. \tag{11}$$

Averaging over all samples yields the $k$-NN graph homophily score:

$$
\begin{aligned}
T_k(\mathcal{Z}) &= \frac{1}{n} \sum_{i=1}^n t_k(i) \\
&= \frac{1}{nk} \sum_{i=1}^n \sum_{j \in \mathcal{N}_k(i)} \mathbb{I}[s_j = s_i].
\end{aligned} \tag{12}
$$

We denote by $T_{\text{null}}$ the expected homophily rate under a random mixing assumption, which accounts for source priors:

$$T_{\text{null}}(\mathcal{Z}) = \frac{\sum_{s=0}^S n_s(n_s - 1)}{n(n-1)}, \tag{13}$$

where $n_s$ is the number of samples from source $s$ in $\mathcal{Z}$. Using this baseline as the zero point, we normalize $T_k(\mathcal{Z})$ as:

$$\widetilde{T}_k(\mathcal{Z}) = \frac{T_k(\mathcal{Z}) - T_{\text{null}}(\mathcal{Z})}{1 - T_{\text{null}}(\mathcal{Z})}. \tag{14}$$

When samples from the same source are more likely to be adjacent within local neighborhoods in the embedding space, the fraction of same-source edges increases and $\widetilde{T}_k$

rises. When local neighborhoods approach random mixing, $T_k \approx T_{\text{null}}$ and thus $\widetilde{T}_k \approx 0$. The derivation of $T_{\text{null}}$ is provided in Appendix B.

## 4. Experiments

### 4.1. Experimental Setup

**Detectors.** We compare against 16 widely used detectors. CNN-based detectors include ResNet-50 (He et al., 2016), CNNDetection (Wang et al., 2020), LGrad (Tan et al., 2023), Gram-Net (Liu et al., 2020), FreqNet (Tan et al., 2024a), NPR (Tan et al., 2024b), and SAFE (Li et al., 2025a). CLIP-based detectors include LASTED (Wu et al., 2025a), UniFD (Ojha et al., 2023), C2P-CLIP (Tan et al., 2025) and our CLIPDetection (Radford et al., 2021) implementation, which attaches a randomly initialized binary classification head to a pretrained CLIP encoder. We also include AIDE (Yan et al., 2024a), which combines CLIP and ResNet-50, DIRE (Wang et al., 2023a), which uses diffusion reconstruction errors for detection, Effort (Yan et al., 2024b), which improves generalization via orthogonal subspace decomposition, SPAI (Karageorgiou et al., 2025), which learns spectral representations for arbitrary-resolution detection, and LOTA (Wang et al., 2025), which exploits noise patterns in low-order bit planes. More details on these methods and their implementations appear in Appendix A.

**Experiment Details.** We fine-tune DINOv3 ViT-L/16[1] (Siméoni et al., 2025) end-to-end using AdamW (Loshchilov & Hutter, 2017), updating all model parameters, with weight decay set to $1 \times 10^{-4}$ and AdamW coefficients $\beta_1 = 0.9$ and $\beta_2 = 0.99$. We apply cosine learning-rate decay with an initial learning rate of $3 \times 10^{-5}$ and a 2000-step linear warm-up. Training runs for 20 epochs on $4\times$ NVIDIA A100 80GB GPUs with bf16 mixed precision. The input resolution is $224 \times 224$, and the global batch size is 1280. The contrastive-learning temperature $\tau$ is set to 0.07, and we set $\lambda = 1$ in Equation 7.

During inference, we use both non-parametric and parametric classifiers. The non-parametric methods include $k$-NN (Cover & Hart, 1967) and prototypical classification, where the retrieval set is the training set or the few-shot support set; prototypes are computed as class-wise mean features. Both are implemented with Faiss-GPU (Meta, 2024). The parametric methods include a linear head trained on the few-shot support set with AdamW using sigmoid outputs, and an RBF-kernel SVM (Cortes & Vapnik, 1995) fitted with cuML (Raschka et al., 2020).

**Datasets and Metrics.** We evaluate on five challenging benchmarks: WildFake (Hong & Zhang, 2024), Commu-

---

[1]facebook/dinov3-vitl16-pretrain-lvd1689m

*Table 2.* **In-domain and cross-dataset detection performance comparison.** All methods are trained only on WildFake. We report results on WildFake (ID), and test directly on Community / AIGIBench / Chameleon / GenImage (OOD) without any retraining. Metrics are AUROC and TPR@FPR=5% (TPR5%). Average is the average over all datasets. Ours estimates a decision boundary via $k$-NN on WildFake-trained features; fine- and coarse-grained ablations are listed under Ours. Best results are **bolded**.

| Method | WildFake | | Community | | AIGIBench | | Chameleon | | GenImage | | Average | |
|---|---|---|---|---|---|---|---|---|---|---|---|---|
| | AUROC | TPR5% | AUROC | TPR5% | AUROC | TPR5% | AUROC | TPR5% | AUROC | TPR5% | AUROC | TPR5% |
| ResNet-50 | 99.71 | 99.59 | 83.62 | 42.10 | 69.70 | 28.81 | 63.65 | 5.77 | 91.49 | 60.31 | 81.63 | 47.32 |
| CNNDetection | 86.98 | 37.52 | 67.62 | 13.68 | 54.36 | 8.93 | 56.04 | 5.61 | 43.96 | 0.61 | 61.79 | 13.27 |
| LGrad | 99.29 | 97.29 | 86.48 | 56.22 | 65.61 | 24.91 | 74.23 | 11.34 | 99.29 | 97.31 | 84.98 | 57.41 |
| Gram-Net | 99.61 | 98.82 | 81.18 | 38.54 | 65.87 | 26.82 | 67.77 | 9.78 | 90.50 | 53.37 | 80.99 | 45.47 |
| FreqNet | 96.40 | 82.29 | 74.80 | 29.29 | 54.58 | 7.30 | 58.14 | 7.76 | 66.09 | 12.68 | 70.00 | 27.86 |
| NPR | 74.27 | 14.72 | 56.25 | 10.51 | 42.15 | 5.81 | 68.96 | 22.77 | 60.41 | 4.29 | 60.41 | 11.62 |
| SAFE | 98.22 | 92.77 | 83.27 | 45.37 | 68.61 | 19.26 | 67.73 | 14.36 | 94.22 | 77.50 | 82.41 | 49.85 |
| LASTED | 97.31 | 85.08 | 73.92 | 25.06 | 57.75 | 15.55 | 61.94 | 10.68 | 85.26 | 23.35 | 75.24 | 31.94 |
| UniFD | 96.52 | 81.40 | 87.63 | 50.92 | 86.67 | 36.57 | 62.66 | 8.57 | 80.76 | 33.99 | 82.85 | 42.29 |
| CLIPDetection | 99.98 | **99.99** | 94.19 | 72.17 | 73.01 | 39.30 | 60.81 | 0.36 | 97.74 | 91.28 | 85.15 | 60.62 |
| LOTA | 68.22 | 20.29 | 56.88 | 16.52 | 61.06 | 8.33 | 57.67 | 9.89 | 67.61 | 16.87 | 62.29 | 14.38 |
| AIDE | 99.22 | 97.91 | 91.75 | 66.85 | 63.48 | 29.62 | 56.45 | 2.64 | 95.60 | 88.94 | 81.30 | 57.19 |
| SPAI | 93.74 | 67.89 | 88.05 | 45.79 | 72.10 | 12.78 | 78.89 | 34.41 | 84.10 | 30.24 | 83.38 | 38.22 |
| C2P-CLIP | 99.99 | 99.98 | 89.41 | 82.01 | 85.85 | 49.89 | 64.18 | 32.54 | 94.71 | 94.69 | 86.83 | 71.82 |
| Effort | 99.99 | **99.99** | 88.51 | 71.78 | 87.91 | 52.90 | 61.69 | 8.05 | 94.02 | 92.39 | 86.42 | 65.02 |
| DIRE | 99.98 | 99.98 | 90.02 | 53.19 | 71.11 | 42.19 | 81.35 | 25.18 | 97.59 | 87.72 | 88.01 | 61.65 |
| **Ours** | **99.99** | 99.98 | **96.95** | **87.50** | **98.00** | **93.75** | **96.35** | **86.38** | **99.84** | **99.55** | **98.23** | **93.43** |
| *w/o fine-grained* | 100.00 | 99.99 | 95.61 | 80.44 | 88.83 | 37.64 | 90.08 | 70.01 | 99.24 | 97.45 | 94.75 | 77.11 |
| *w/o coarse-grained* | 99.93 | 99.92 | 90.43 | 48.98 | 93.15 | 67.54 | 88.00 | 38.89 | 96.71 | 82.62 | 93.64 | 67.59 |

nity Forensics (Community) (Park & Owens, 2025), AI-GIBench (Li et al., 2025b), Chameleon (Yan et al., 2024a), and GenImage (Zhu et al., 2023).

We report AUROC and TPR at 5% false positive rate (TPR@FPR=5%, denoted as TPR5%), measured at the threshold where the false positive rate is fixed to 5%, which reflects performance at a low false-alarm operating point. All results are reported in percentage points and we omit the percent sign for brevity (e.g., 99.9 denotes 99.9%).

### 4.2. Cross-Dataset Generalization

Given the high in-domain (ID) performance across methods, we evaluate generalization under out-of-distribution (OOD) shifts. Under a unified protocol, all methods are trained only on WildFake and directly evaluated on Community, AI-GIBench, Chameleon, and GenImage without any retraining. For reproducibility, we strictly align preprocessing across methods, following official settings or the AIGIBench implementation. Besides training on WildFake, we also report results when training on Community in Appendix E. Fine- and coarse-grained ablations are reported under Ours.

Table 2 shows that (i) on WildFake (ID), most methods already achieve high AUROC (e.g., ResNet-50, Gram-Net, LGrad, SAFE, CLIPDetection, AIDE, DIRE, and FiSeR reach 98–99.99), indicating ID performance is close to the ceiling; (ii) under cross-dataset testing, most baselines de-

*Table 3.* **OOD generalization comparison across different vision backbones.** All evaluations are conducted on OOD test sets using WildFake as the source domain. "Pretrained" denotes using the off-the-shelf backbone without updating its parameters on WildFake, where the WildFake training split is used only to build the feature bank for inference. "Fine-tuned" denotes updating the backbone on the WildFake training split before direct evaluation on OOD benchmarks. Metrics are AUROC and TPR@FPR=5%.

| Backbone | Setting | AIGIBench | | Chameleon | |
|---|---|---|---|---|---|
| | | AUROC | TPR5% | AUROC | TPR5% |
| DINOv3 ViT-L/16 | Pretrained | 89.87 | 59.23 | 85.42 | 57.91 |
| | Fine-tuned | 98.00 | 93.75 | 96.35 | 86.38 |
| DINOv3 ViT-H/16+ | Pretrained | 92.02 | 71.52 | 81.91 | 46.35 |
| | Fine-tuned | 97.24 | 90.41 | 94.52 | 80.53 |
| CLIP ViT-L/14 | Pretrained | 84.91 | 29.71 | 79.07 | 50.34 |
| | Fine-tuned | 93.78 | 74.53 | 85.08 | 56.61 |
| DFN2B-CLIP ViT-L/14 | Pretrained | 71.22 | 1.18 | 60.11 | 36.38 |
| | Fine-tuned | 92.35 | 64.10 | 75.90 | 28.98 |
| DINOv2-Large | Pretrained | 85.49 | 42.47 | 80.81 | 45.73 |
| | Fine-tuned | 95.12 | 70.52 | 92.82 | 74.99 |

grade substantially, with the largest drops on the most challenging AIGIBench and Chameleon; (iii) in contrast, FiSeR remains stable on these hard datasets, achieving 98.00/96.35 AUROC on AIGIBench/Chameleon, and obtains the best Average score of 98.23 AUROC and 93.43 TPR5%, outperforming the second-best DIRE by +10.22 AUROC and +31.78 TPR5%; (iv) ablations indicate that the fine-grained loss is critical for cross-domain generalization: using only the fine-grained loss and training with synthetic images only,

*Table 4.* **Comparison of classification heads.** We freeze representations trained on WildFake and evaluate four few-shot classifiers on AIGIBench using 10-shot labeled target samples. We report AUROC (mean±std) over 5 random few-shot draws. Average is the mean AUROC of each classifier averaged across all methods.

| Method | $k$-NN | Linear | SVM | Proto |
|---|---|---|---|---|
| ResNet-50 | 74.13±3.43 | 59.07±12.09 | **77.53±2.85** | 74.96±4.28 |
| CNNDetection | 71.57±9.12 | **90.83±1.03** | 85.77±2.29 | 72.79±1.81 |
| CLIPDetection | 83.99±2.12 | **86.06±1.08** | 78.86±1.63 | 79.17±0.20 |
| UniFD | 87.63±3.44 | 89.77±2.77 | **90.42±1.89** | 86.61±3.48 |
| DIRE | 88.21±2.21 | **91.71±0.64** | 86.99±5.47 | 87.35±0.91 |
| Ours | 99.19±0.50 | 98.97±0.37 | **99.23±0.17** | 98.64±0.37 |
| Average | 84.12 | 86.07 | **86.47** | 83.25 |

the model still performs well for synthetic image detection at test time (last row in Table 2). This suggests that fine-grained supervision encourages transferable representations tied to generation mechanisms and domain shifts, enabling FiSeR to distinguish natural and synthetic images even when the natural domain is unseen during training. Meanwhile, jointly supervising with the coarse-grained loss yields the best performance.

We further examine the role of backbone choice in Table 3. Different pretrained backbones exhibit noticeably different OOD behavior. Fine-tuning on WildFake substantially improves AUROC across all backbones and both OOD benchmarks, and generally improves TPR@FPR=5% in most cases. These results suggest that the gains are not merely inherited from off-the-shelf pretrained features. Moreover, larger backbones do not necessarily lead to stronger OOD performance, as DINOv3 ViT-L/16 outperforms DINOv3 ViT-H/16+ after fine-tuning. These results indicate that FiSeR's robustness depends not only on backbone choice, but also on its hierarchical supervised contrastive representation learning.

### 4.3. Feature Separability and Few-Shot Cross-Domain Generalization

In Section 4.2, we observed a significant performance drop when detectors are tested across datasets. To verify whether this degradation mainly comes from a mismatch of the classification boundary in the head rather than a failure of intermediate representations, we freeze the backbone and refit a lightweight classification head using a small number of labeled samples from the target domain.

As shown in Table 1, under 10-shot supervision, an SVM head improves CLIP-Detection from 60.81 to 85.39 AUROC and ResNet50-Detection from 56.04 to 92.38. To test whether this trend holds more broadly, we further evaluate few-shot SVM refitting for all methods in an OOD setting. Figure 3 reports few-shot SVM refitting under OOD settings (training on WildFake/Community and testing on AIGIBench/Chameleon).

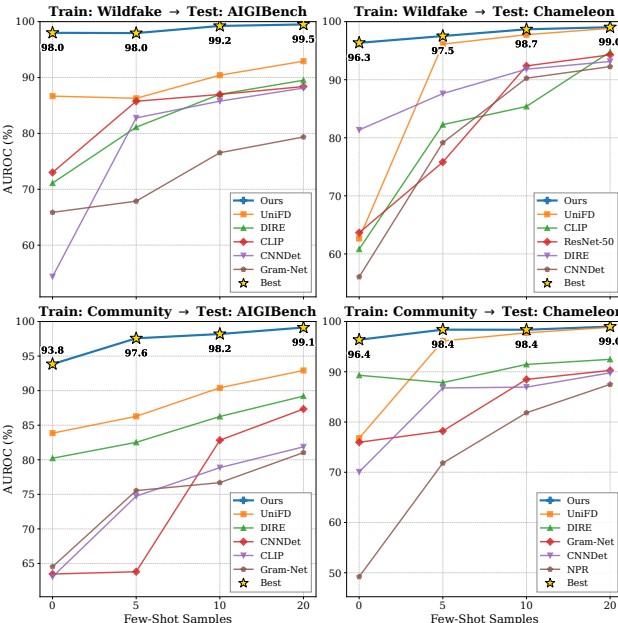

*Figure 3.* **Few-shot SVM refitting on OOD domains.** We report our method and the top-5 baselines. For each method, we select the intermediate layer with the highest AUROC. On each OOD domain, we train an SVM with $N$ shots per class and report AUROC averaged over 5 random draws. Stars indicate the best-performing method at each $N$.

All results for the detectors are reported in Appendix F. As the number of shots increases from 5 to 20, AUROC improves steadily for all methods, with a particularly noticeable gain already at 5 shots. This suggests that cross-domain performance degradation mainly comes from a mismatch between the detection head and the target-domain distribution, and that a small number of target-domain samples can yield substantial benefits. FiSeR reaches near-saturated performance with very few target-domain samples and improves AUROC to 99 at 20 shots.

To verify that the gain is not tied to a specific classifier, we compare multiple classifier heads in Table 4, including $k$-NN, a linear classifier, SVM, and a prototypical classifier. All methods show consistent and significant improvements under different classification heads. FiSeR remains empirically close to the best performance across all heads, indicating stronger inter-class separability and tighter intra-class compactness. Moreover, the improvement magnitude is similar across heads, further supporting that the gain is largely classifier-agnostic.

To further quantify representation quality, we adopt the $k$-NN graph homophily metric in Section 3.3. As shown in Table 5, the homophily score exhibits an approximately linear alignment with the empirically best few-shot AUROC across these datasets. Here, "empirically best" refers to the best result observed under our evaluation protocol, across a finite number of support-set samplings and the considered

*Table 5.* ***k*-NN graph homophily scores of intermediate layer representations across detectors.** For each detector, we compute homophily for all intermediate layers and report the layer with the highest score. Scores are averaged over $k \in \{1, 5, 10, 15, 20\}$. The upper block trains on WildFake and the lower block trains on Community; both are evaluated on Community / AIGIBench / Chameleon. Higher scores mean that a sample's nearest neighbors more often share the same source (natural vs. synthetic). Best results are **bolded**.

| | ResNet-50 | CNNDetection | LGrad | Gram-Net | FreqNet | NPR | SAFE | LASTED | UniFD | CLIPDetection | AIDE | DIRE | Ours |
|---|---|---|---|---|---|---|---|---|---|---|---|---|---|
| **WildFake Train** | | | | | | | | | | | | | |
| Community | 80.96 | 82.88 | 50.59 | 63.45 | 31.44 | 76.48 | 68.41 | 51.95 | 92.95 | 86.98 | 80.69 | 82.77 | **94.18** |
| Chameleon | 90.60 | 80.93 | 54.07 | 67.95 | 34.56 | 82.21 | 60.11 | 55.47 | **98.06** | 93.78 | 60.28 | 81.17 | 95.50 |
| AIGIBench | 66.86 | 80.65 | 52.09 | 62.34 | 39.13 | 65.06 | 50.81 | 39.18 | 92.14 | 81.62 | 49.60 | 81.78 | **98.45** |
| **Community Train** | | | | | | | | | | | | | |
| Community | 74.01 | 78.52 | 44.31 | 71.50 | 41.32 | 71.28 | 65.99 | 42.65 | 92.95 | 79.01 | 77.54 | 80.56 | **93.87** |
| Chameleon | 87.96 | 77.11 | 47.10 | 77.02 | 40.66 | 72.03 | 66.61 | 48.16 | **98.06** | 76.91 | 40.28 | 82.01 | 96.26 |
| AIGIBench | 62.39 | 77.14 | 48.57 | 65.72 | 44.51 | 64.14 | 49.73 | 22.23 | 92.14 | 74.92 | 50.17 | 79.38 | **98.16** |

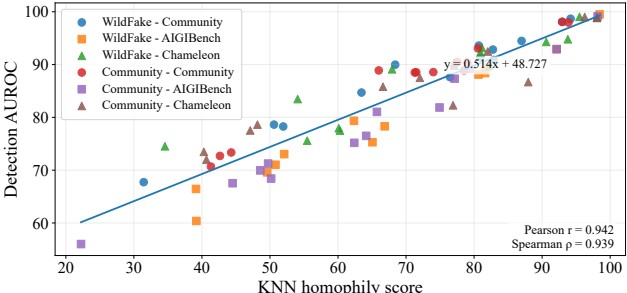

*Figure 4.* **Correlation between $k$-NN graph homophily and few-shot SVM performance.** Each point pairs a detector's 20-shot SVM AUROC with its $k$-NN graph homophily score. We fit a least-squares linear regressor; the legend denotes the train–test domain pair. Pearson $r$ and Spearman $\rho$ are reported in the figure.

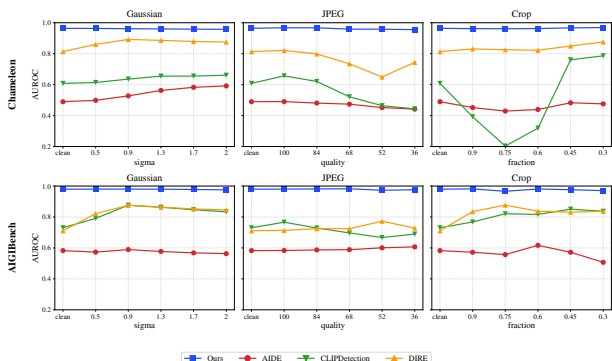

*Figure 5.* **Robustness comparison under common image perturbations.** We report AUROC on the Chameleon and AIGIBench OOD test sets under three perturbation types: Gaussian blur, JPEG compression, and center cropping. Rows correspond to datasets and columns correspond to perturbation types. For each perturbation, severity increases from the clean setting to stronger degradation; for JPEG compression and center cropping, lower quality/fraction values indicate stronger perturbations. Each curve reports the AUROC of FiSeR (Ours), AIDE, CLIPDetection, and DIRE across severity levels.

set of classifier heads. In contrast to few-shot AUROC, which can have large variance due to the number of shots, the classifier head, and support-set sampling, the homophily score can be computed once from the representation for a fixed $k$ and is numerically stable. We therefore use it as a low-variance proxy for representation separability and as an indicator of few-shot performance trends, and conduct a correlation analysis. We provide a Monte Carlo validation of the random-mixing baseline $T_{\text{null}}$ in Appendix D.

As shown in Figure 4, the Pearson correlation coefficient is 0.942 and the Spearman rank correlation coefficient is 0.939, indicating a very strong monotonic relationship with a near-linear trend between the homophily score and few-shot AUROC.

Moreover, when the training set or training domain changes, FiSeR's homophily score varies only slightly, suggesting that its representations remain stable across training distributions and transfer well.

### 4.4. Representation Robustness, Separability, and Transferability

We further evaluate the robustness of FiSeR under common test-time perturbations, including Gaussian blur, JPEG com-

pression, and center cropping. As shown in Figure 5, FiSeR maintains high AUROC on both Chameleon and AIGIBench across the tested corruption severities. In contrast, AIDE, CLIPDetection, and DIRE exhibit larger performance fluctuations and more pronounced degradation under several perturbations, particularly JPEG compression and aggressive cropping. These perturbations either suppress high-frequency details or remove spatial context, suggesting that existing detectors can be more sensitive to local texture and spatial cues. These results indicate that FiSeR learns a more stable representation under common image perturbations, rather than relying primarily on low-level artifacts that are sensitive to post-processing.

To verify whether the representations learned by FiSeR simultaneously capture fine-grained generator identity information and transfer across datasets, we analyze the representations learned on WildFake. Specifically, we use UMAP to project the features into a two-dimensional space. As

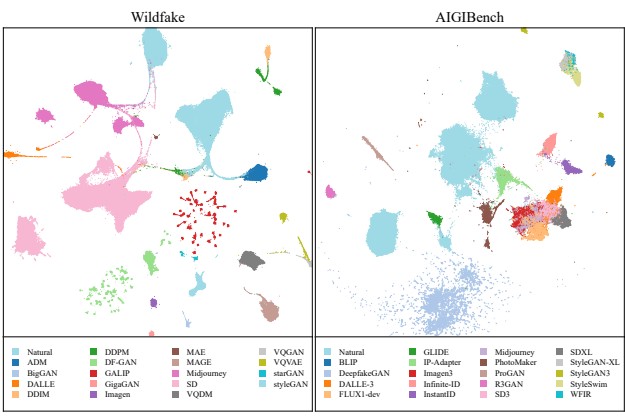

*Figure 6.* **UMAP visualization of FiSeR's representations.** Trained on WildFake, we project extracted features to 2D using unsupervised UMAP for multi-class visualization. Left: WildFake test set (ID). Right: AIGIBench test set (OOD).

shown in Figure 6, on the WildFake (ID) test set, natural images and synthetic images exhibit a relatively stable separation in the feature space; meanwhile, synthetic images further form multiple compact local clusters according to generator categories, indicating that the fine-grained objective preserves generator identity information in the synthetic feature space. We provide the UMAP visualization of the pretrained DINOv3 representations in Appendix C.

On the AIGIBench (OOD) test set, although the data distribution shifts, the separability between natural and synthetic images is maintained, and synthetic samples do not collapse into a single cluster but still show multiple distinguishable clusters. At the same time, natural images also form three clusters in the feature space, consistent with the fact that the natural data in AIGIBench come from three domains.

Furthermore, we compute the homophily score (higher indicates that the two classes are more separable) between generators on AIGIBench. As shown in Figure 7, most generator pairs exhibit high separability; the relatively low-separability regions are mainly concentrated among model families with more similar generation mechanisms, such as some StyleGAN (Karras et al., 2019) variants, consistent with the expectation that they have more similar visual statistics and are harder to distinguish.

## 5. Conclusion

We study cross-domain generalization for AI-generated image detection under real distribution shift and identify a key bottleneck behind performance degradation. Experiments show that even when the target domain differs substantially from the source, freezing the backbone and lightly adapting only the classification head with a few labeled target examples, using a linear probe, SVM, or $k$-NN, can yield clear gains. This suggests that the learned representations already contain transferable discriminative information, and

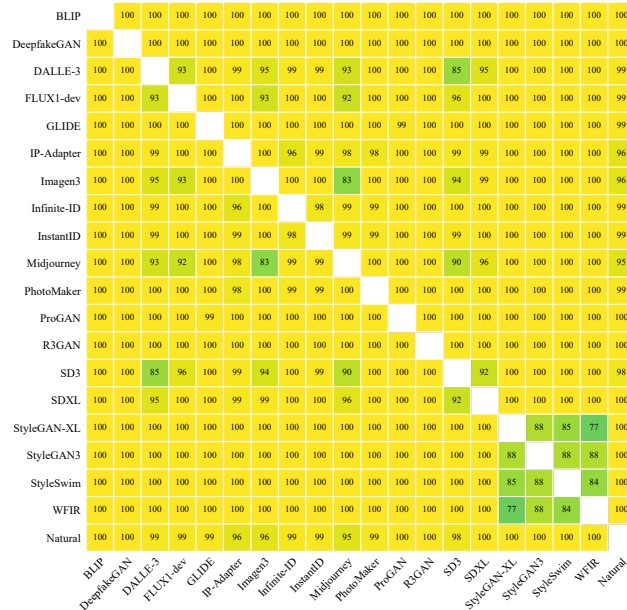

*Figure 7.* **Pairwise class separability heatmap on AIGIBench.** We train FiSeR on WildFake, compute pairwise homophily scores between classes on the AIGIBench test set; higher scores indicate stronger separability in the learned feature space.

that the drop in cross-domain performance is more likely caused by a source-trained classification head whose decision boundary is biased on the new domain. Motivated by this observation, we propose a hierarchical contrastive learning framework that turns the discriminative structure in intermediate features into more transferable and stable representations, leading to state-of-the-art performance across multiple cross-domain benchmarks. Our results also indicate that lightweight few-shot adaptation with limited target samples can substantially improve robustness under domain shift, and may be informative for detection tasks in other modalities and distribution-shift settings.

## Impact Statement

This work aims to improve the robustness of cross-domain synthetic image detection, with potential applications in synthetic content provenance, platform governance, and benchmark comparison. At the same time, stronger detection may accelerate evasion, and false positives may lead to inappropriate actions and reputational harm. To mitigate these risks, we recommend output calibration and human review, and caution against using a single model prediction as the sole basis for high-stakes decisions.

## Acknowledgements

This work was supported by the National Key R&D Program of China (Grant No. 2024YFC3210804).

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

# A. Baselines and Implemented Methods

Unless stated otherwise, we train models end to end for natural vs. synthetic binary classification with an input resolution of $224 \times 224$, and use the data augmentation pipeline from AIGI (Li et al., 2025b).

**ResNet-50.** We use ResNet-50 (He et al., 2016) as a comparison baseline. The network is randomly initialized and trained end to end as a natural vs. synthetic binary classifier with an input resolution of $224 \times 224$.

**CNNDetection.** CNNDetection (Wang et al., 2020) formulates detection as a real vs. generated binary classification task. It trains a ResNet-50 on real images and images synthesized by a specific generator (e.g., ProGAN), and improves generalization to unseen architectures and domains via carefully designed pre and post processing and augmentations that mimic common post-processing operations.

**LGrad.** LGrad (Tan et al., 2023) uses a fixed pretrained ResNet-50 as a transformation model to convert an image into gradients. These gradients are treated as a generic artifact representation and fed into a classifier to predict authenticity.

**Gram-Net.** Gram-Net (Liu et al., 2020) explicitly introduces Gram matrix modules into ResNet-18. A Gram block is attached at the input and placed before each downsampling stage to inject global texture information across semantic levels. Each block first aligns feature dimensions with a convolution, then extracts global texture features via a Gram matrix layer, refines them with two conv-bn-relu layers, matches the backbone scale through global pooling, and finally concatenates the resulting features with the ResNet backbone for binary classification.

**FreqNet.** FreqNet (Tan et al., 2024a) is centered on frequency-domain learning. It first applies FFT to the input image and uses a high-pass filter to retain only high-frequency components; the inverse transform produces a high-frequency image that is fed to the detector. Inside the network, it further performs frequency transforms and high-pass filtering over intermediate features along spatial and channel dimensions to obtain high-frequency feature representations, implemented as a plug-and-play module combined with convolutional residual blocks. The model is trained with standard binary cross-entropy for real vs. fake classification. In our implementation, we use the lightweight CNN classifier designed by the authors, based on residual convolution blocks, trained from scratch at $224 \times 224$ without pretraining.

**NPR (ResNet-1.44M).** NPR (Tan et al., 2024b) observes that upsampling and subsequent convolutions in generative models introduce observable local pixel dependencies in generated images. It proposes neighborhood pixel relationships (NPR) to capture and represent common structural artifacts induced by upsampling. The NPR representation is then used to train a binary detector to distinguish real and generated images.

**SAFE (ResNet-1.44M).** SAFE (Li et al., 2025a) introduces three simple image transformations. It replaces downsampling with a cropping operator during preprocessing, and applies color jitter and random rotation, forcing the detector to focus on local regions during training.

**LASTED.** LASTED (Wu et al., 2025a) trains with carefully designed text-augmented labels, such as "A photo/painting of [Image Captioning]", and uses contrastive vision-language supervision to obtain a CLIP feature space with improved generalization for forensics. After training, it freezes the image encoder and trains a linear classification head with binary cross-entropy for natural vs. synthetic classification.

**UniFD.** UniFD (Ojha et al., 2023) uses a large pretrained CLIP visual encoder as a feature extractor. It builds a feature bank from real and fake samples in the training set; at test time, it extracts features and performs nearest-neighbor matching using cosine distance. It also provides a linear-probe variant, which freezes the feature extractor and trains a single-layer linear classifier with binary cross-entropy. In our implementation, we use $224 \times 224$ inputs and adopt the linear-probe setting with a single-layer classifier.

**CLIPDetection.** CLIPDetection trains a randomly initialized binary classification head on top of CLIP embeddings, while updating both the CLIP backbone and the head.

**AIDE.**  AIDE (Yan et al., 2024a) is a hybrid-feature detector that uses multiple experts to extract visual artifacts and noise patterns. To capture high-level semantics, it computes CLIP visual embeddings. It then splits the image into $32 \times 32$ patches, selects patches with the highest and lowest frequencies, and extracts patch features using ResNet-50. The expert features are fused and fed into an MLP with linear layers and GELU to produce the final score.

**DIRE.**  DIRE (Wang et al., 2023a) proposes a representation based on diffusion reconstruction error. Given an input image, it performs DDIM inversion with a pretrained diffusion model to progressively add noise, then denoises to obtain a residual image. The pixel-wise absolute difference between the input and the residual image is used as the DIRE feature. A binary classifier is trained on DIRE features with binary cross-entropy; inference follows the same pipeline by computing DIRE and then applying the classifier.

**Effort.**  Effort (Yan et al., 2024b) improves the generalization of AI-generated image detection via orthogonal subspace decomposition. It uses SVD to divide the feature space of a pretrained visual model into semantic and forgery-related subspaces, freezes the dominant semantic components, and learns forgery cues only in the residual subspace, thereby reducing overfitting to generator-specific artifacts.

**SPAI.**  SPAI (Karageorgiou et al., 2025) detects AI-generated images from a spectral perspective, assuming that real images exhibit more stable frequency distributions. It learns real-image spectral patterns through masked spectral learning and compares representations of the original, low-frequency, and high-frequency images. It also supports arbitrary-resolution inputs by aggregating patch-level spectral features for final classification.

**C2P-CLIP.**  C2P-CLIP (Tan et al., 2025) builds on CLIP to improve generalization in deepfake detection. It introduces category-common prompts for real and fake classes, such as "camera" for real images and "deepfake" for fake images, and trains with both image-text contrastive learning and image classification losses. The backbone is kept frozen, while lightweight adaptation is performed with LoRA.

**LOTA.**  LOTA (Wang et al., 2025) detects AI-generated images using noise patterns in low-order bit-planes. It decomposes an image into bit-planes, constructs a noise image from the low-order planes, selects the most discriminative patches based on gradient responses, and feeds them into a classifier for real/fake prediction.

## B. $T_{\text{null}}$ under the Random Mixing Assumption

Let $\mathcal{Z} = \{(z_i, s_i)\}_{i=1}^{n}$, where $s_i \in \{0, \dots, S\}$ denotes the source label of sample $i$. Let $n_s$ be the number of samples from source $s$, so that $\sum_{s=0}^{S} n_s = n$.

For sample $x_i$, the proportion of same-source neighbors is

$$t_k(i) = \frac{1}{k} \sum_{j \in \mathcal{N}_k(i)} \mathbb{I}[s_j = s_i],$$

and the $k$-NN graph homophily score is

$$T_k(\mathcal{Z}) = \frac{1}{n} \sum_{i=1}^{n} t_k(i).$$

We define $T_{\text{null}}(\mathcal{Z})$ as the expected homophily rate under the random mixing assumption: for each $i$, the neighbor set $\mathcal{N}_k(i)$ is obtained by uniformly sampling $k$ distinct samples from the remaining $n - 1$ samples, independently of the source labels.

**Proposition.**  Under the above assumption,

$$T_{\text{null}}(\mathcal{Z}) = \frac{\sum_{s=0}^{S} n_s(n_s - 1)}{n(n - 1)}.$$

**Proof.**  Fix a sample $i$ and suppose $s_i = s$. Under random mixing, consider any neighbor position. Since the $k$ neighbors are sampled without replacement and the sampling procedure is symmetric, the selected sample at this position is uniformly

distributed over the $n-1$ candidates. Therefore, the probability that this neighbor has the same source $s$ equals the fraction of source-$s$ samples among the $n-1$ candidates:

$$\mathbb{P}(s_j = s_i \mid s_i = s) = \frac{n_s - 1}{n - 1}.$$

It follows that

$$\mathbb{E}[t_k(i) \mid s_i = s] = \mathbb{E}\left[\frac{1}{k} \sum_{j \in \mathcal{N}_k(i)} \mathbb{I}[s_j = s_i] \;\middle|\; s_i = s\right] = \frac{1}{k} \sum_{j=1}^{k} \mathbb{P}(s_j = s_i \mid s_i = s) = \frac{n_s - 1}{n - 1}.$$

Averaging over all samples yields

$$\mathbb{E}[T_k(\mathcal{Z})] = \frac{1}{n} \sum_{i=1}^{n} \mathbb{E}[t_k(i)] = \sum_{s=0}^{S} \frac{n_s}{n} \cdot \frac{n_s - 1}{n - 1} = \frac{\sum_{s=0}^{S} n_s(n_s - 1)}{n(n - 1)}.$$

By definition, this quantity is $T_{\text{null}}(\mathcal{Z})$.

## C. Representation Visualization.

We further analyze the learned representations using UMAP visualization. As shown in Figure 8, compared with the pretrained DINOv3 features, FiSeR produces a clearer class structure on the WildFake test set, separating natural images and synthetic samples from different generators into more distinguishable clusters. This suggests that, after training on WildFake, FiSeR learns generation-aware representations beyond the generic semantic structure of the pretrained backbone.

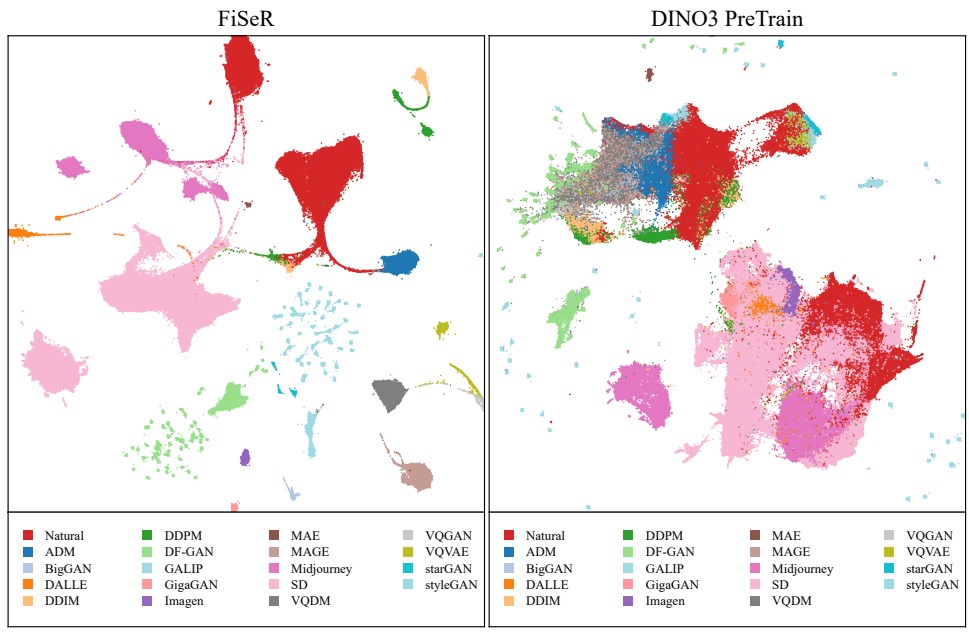

*Figure 8.* **UMAP visualization of representations on the WildFake test set.** We project features extracted from the WildFake test set into 2D using unsupervised UMAP for multi-class visualization. The left panel shows FiSeR representations learned on WildFake train, while the right panel shows DINOv3 ViT-L/16 pretrained representations without training.

## D. Monte Carlo Validation of $T_{\text{null}}$

Table 6 validates the theoretical null expectation $T_{\text{null}}$ through Monte Carlo simulations under diverse 2D geometric structures. We vary the source-count configuration, the geometry template, and the neighborhood size $k$, and estimate empirical homophily by repeatedly permuting source labels under fixed source counts. Across all settings, the empirical means closely match the theoretical $T_{\text{null}}$ values, with only small absolute errors. These results confirm that $T_{\text{null}}$ provides an accurate and geometry-agnostic baseline for homophily analysis.

*Table 6.* **Monte Carlo validation of $T_{\text{null}}$ under predefined 2D geometric structures with mixed $k$.** We evaluate 36 settings in total, formed by 9 source-count configurations and 4 predefined 2D geometry templates. "Shape" denotes the predefined geometric patterns used to generate the fixed 2D point cloud, including Gaussian, Ring, Moon, Spiral, and Line; "+" indicates that multiple patterns are combined in the same geometry template. $k$ denotes the number of neighbors used to construct the $k$-NN graph. For each setting, we first fix a geometry template and generate 24 random instances from that template; then, while keeping the source counts unchanged, we perform 200 random label permutations for each instance to estimate the mean empirical homophily under the random-mixing null. Therefore, each row is based on 4800 Monte Carlo samples in total. The table reports the theoretical $T_{\text{null}}$, the empirical mean, and their absolute error.

| Source counts | Shape | $k$ | Theoretical $T_{\text{null}}$ | Empirical mean | Abs. error |
|---|---|---|---|---|---|
| [60, 60] | Gaussian + Ring | 20 | 0.495798 | 0.495799 | 2.05e-07 |
| [60, 60] | Moon + Spiral | 16 | 0.495798 | 0.496137 | 3.38e-04 |
| [60, 60] | Line + Gaussian | 11 | 0.495798 | 0.495822 | 2.37e-05 |
| [60, 60] | Ring + Line | 6 | 0.495798 | 0.496104 | 3.05e-04 |
| [40, 80] | Gaussian + Ring | 9 | 0.551821 | 0.551606 | 2.15e-04 |
| [40, 80] | Moon + Spiral | 16 | 0.551821 | 0.551671 | 1.50e-04 |
| [40, 80] | Line + Gaussian | 5 | 0.551821 | 0.551663 | 1.58e-04 |
| [40, 80] | Ring + Line | 14 | 0.551821 | 0.552113 | 2.92e-04 |
| [20, 100] | Gaussian + Ring | 15 | 0.719888 | 0.719968 | 8.04e-05 |
| [20, 100] | Moon + Spiral | 13 | 0.719888 | 0.719890 | 2.27e-06 |
| [20, 100] | Line + Gaussian | 20 | 0.719888 | 0.719716 | 1.72e-04 |
| [20, 100] | Ring + Line | 4 | 0.719888 | 0.719685 | 2.03e-04 |
| [40, 40, 40] | Gaussian + Ring + Moon | 17 | 0.327731 | 0.327673 | 5.77e-05 |
| [40, 40, 40] | Spiral + Gaussian + Line | 7 | 0.327731 | 0.326608 | 1.12e-03 |
| [40, 40, 40] | Moon + Ring + Spiral | 18 | 0.327731 | 0.327673 | 5.78e-05 |
| [40, 40, 40] | Line + Moon + Gaussian | 4 | 0.327731 | 0.327432 | 2.99e-04 |
| [20, 40, 60] | Gaussian + Ring + Moon | 18 | 0.383754 | 0.383758 | 4.99e-06 |
| [20, 40, 60] | Spiral + Gaussian + Line | 11 | 0.383754 | 0.383521 | 2.33e-04 |
| [20, 40, 60] | Moon + Ring + Spiral | 13 | 0.383754 | 0.384081 | 3.28e-04 |
| [20, 40, 60] | Line + Moon + Gaussian | 12 | 0.383754 | 0.384023 | 2.69e-04 |
| [10, 30, 80] | Gaussian + Ring + Moon | 13 | 0.509804 | 0.509997 | 1.93e-04 |
| [10, 30, 80] | Spiral + Gaussian + Line | 2 | 0.509804 | 0.509471 | 3.33e-04 |
| [10, 30, 80] | Moon + Ring + Spiral | 14 | 0.509804 | 0.509490 | 3.13e-04 |
| [10, 30, 80] | Line + Moon + Gaussian | 9 | 0.509804 | 0.509260 | 5.44e-04 |
| [30, 30, 30, 30] | Gaussian + Ring + Moon + Spiral | 12 | 0.243697 | 0.243638 | 5.99e-05 |
| [30, 30, 30, 30] | Line + Gaussian + Moon + Ring | 16 | 0.243697 | 0.243463 | 2.35e-04 |
| [30, 30, 30, 30] | Spiral + Line + Gaussian + Moon | 17 | 0.243697 | 0.243825 | 1.28e-04 |
| [30, 30, 30, 30] | Ring + Spiral + Line + Gaussian | 6 | 0.243697 | 0.243461 | 2.36e-04 |
| [15, 25, 35, 45] | Gaussian + Ring + Moon + Spiral | 19 | 0.278711 | 0.278922 | 2.11e-04 |
| [15, 25, 35, 45] | Line + Gaussian + Moon + Ring | 7 | 0.278711 | 0.278941 | 2.29e-04 |
| [15, 25, 35, 45] | Spiral + Line + Gaussian + Moon | 11 | 0.278711 | 0.278821 | 1.09e-04 |
| [15, 25, 35, 45] | Ring + Spiral + Line + Gaussian | 9 | 0.278711 | 0.278415 | 2.96e-04 |
| [10, 20, 30, 60] | Gaussian + Ring + Moon + Spiral | 20 | 0.341737 | 0.341766 | 2.94e-05 |
| [10, 20, 30, 60] | Line + Gaussian + Moon + Ring | 15 | 0.341737 | 0.341698 | 3.90e-05 |
| [10, 20, 30, 60] | Spiral + Line + Gaussian + Moon | 1 | 0.341737 | 0.340576 | 1.16e-03 |
| [10, 20, 30, 60] | Ring + Spiral + Line + Gaussian | 7 | 0.341737 | 0.342064 | 3.27e-04 |

# E. Additional Cross-Dataset Generalization Results

This section further reports results where all methods are trained only on Community and directly evaluated on WildFake, AIGIBench, Chameleon, and GenImage without any additional training. We report AUROC and TPR@FPR=5%, together with the average across datasets, as shown in Table 7. Our method estimates the decision boundary via $k$-NN on features trained on Community.

When trained on Community, the overall performance of all methods is generally lower than that of training on WildFake, and most baselines degrade more substantially. For example, ResNet-50 and CLIPDetection drop to 0 TPR5% on Chameleon. In contrast, our method maintains higher and more balanced AUROC and TPR5% across target domains and achieves the best average performance. These results indicate that our approach remains more stable under cross-domain transfer when the source domain is switched to Community, with more pronounced gains on challenging target domains such as AIGIBench and Chameleon.

*Table 7.* **In-domain and cross-dataset detection performance comparison.** All methods are trained only on Community. We report results on Community, and test directly on WildFake / AIGIBench / Chameleon / GenImage without any retraining. Metrics are AUROC and TPR@FPR=5% (TPR5%). Average is the average over all datasets. Ours estimates a decision boundary via $k$-NN on Community-trained features; Best results are **bolded**.

| Method | WildFake | | Community | | AIGIBench | | Chameleon | | GenImage | | Average | |
|---|---|---|---|---|---|---|---|---|---|---|---|---|
| | AUROC | TPR5% | AUROC | TPR5% | AUROC | TPR5% | AUROC | TPR5% | AUROC | TPR5% | AUROC | TPR5% |
| ResNet-50 | 80.34 | 19.88 | 80.49 | 35.49 | 61.12 | 26.31 | 61.15 | 0.00 | 98.86 | 93.46 | 76.39 | 35.03 |
| CNNDetection | 84.07 | 47.49 | 65.32 | 15.11 | 63.47 | 11.87 | 70.07 | 15.14 | 71.63 | 1.71 | 70.91 | 18.26 |
| LGrad | 84.68 | 4.39 | 86.01 | 57.35 | 66.81 | 25.71 | 59.66 | 0.00 | 99.30 | **97.54** | 79.29 | 37.00 |
| Gram-Net | 84.51 | 23.81 | 81.24 | 28.54 | 64.53 | 32.10 | 75.97 | 19.98 | 98.29 | 89.40 | 80.91 | 38.77 |
| FreqNet | 68.04 | 9.03 | **89.53** | 53.76 | 59.41 | 11.90 | 51.98 | 10.33 | 89.53 | 53.76 | 71.70 | 27.76 |
| NPR | 43.59 | 4.29 | 55.65 | 8.40 | 38.66 | 2.24 | 49.21 | 6.08 | 25.91 | 2.27 | 42.60 | 4.66 |
| SAFE | 83.26 | 46.63 | 73.71 | 24.33 | 65.22 | 20.89 | 78.77 | 31.42 | 95.47 | 84.21 | 79.29 | 41.50 |
| LASTED | 71.10 | 5.38 | 68.33 | 32.53 | 56.43 | 11.50 | 76.34 | 49.98 | 96.46 | 84.71 | 73.73 | 36.82 |
| UniFD | 78.35 | 32.97 | 71.57 | 42.03 | 83.85 | 49.99 | 76.78 | 34.52 | 83.65 | 42.86 | 78.84 | 40.47 |
| CLIPDetection | 83.46 | 0.00 | 88.92 | 42.83 | 63.07 | 29.93 | 63.06 | 0.00 | 98.42 | 93.91 | 79.39 | 33.33 |
| AIDE | 80.45 | 18.83 | 81.27 | 29.50 | 64.60 | 10.54 | 54.89 | 11.08 | **99.43** | 96.82 | 76.13 | 33.35 |
| DIRE | **84.93** | 29.49 | 87.09 | 27.74 | 80.22 | 44.65 | 89.30 | 48.48 | 97.38 | 83.67 | 87.78 | 46.81 |
| **Ours** | 79.73 | **72.30** | 86.99 | **58.55** | **93.84** | **78.33** | **96.37** | **81.47** | 98.69 | 95.95 | **91.12** | **77.32** |

## F. Detailed Few-shot SVM Refitting Results on OOD Domains

This section reports detailed few-shot adaptation results on OOD domains when the training domain is WildFake or Community, as shown in Tables 8 and 9.

The overall trend is consistent with Figure 3. As the number of shots increases from 5 to 20, AUROC improves for all methods, and some methods already obtain substantial gains at 5-shot. This suggests that cross-domain degradation mainly comes from mismatch between the classifier head and the target-domain distribution, and a small amount of target-domain supervision can yield clear benefits. Meanwhile, our method saturates with very few samples and reaches around 99 AUROC at 20-shot. Although our method already achieves strong 0-shot AUROC on OOD domains, SVM refitting further improves performance with small variance. In contrast, some baselines exhibit large variance in the few-shot setting; for example, CNNDetection obtains $69.48 \pm 26.53$ at 5-shot on Community, indicating higher sensitivity to the support-set sampling in their features or adaptation procedure.

*Table 8.* **Detailed few-shot SVM refitting results on OOD domains trained on WildFake.** All methods are trained only on WildFake. We evaluate on three OOD test sets, Community, AIGIBench, and Chameleon. For each method, we select the intermediate layer with the best AUROC, and then train an SVM on the target domain using $N$ labeled samples per class, where $N \in \{0, 5, 10, 20\}$. The metric is AUROC (%); 0-shot denotes using the method's original classifier directly. For each $N$, results are averaged over 5 random draws and reported as mean±std. Best results for each dataset and each $N$ are **bolded**.

| Method | Community | | | | AIGIBench | | | | Chameleon | | | |
|---|---|---|---|---|---|---|---|---|---|---|---|---|
| | 0-shot | 5-shot | 10-shot | 20-shot | 0-shot | 5-shot | 10-shot | 20-shot | 0-shot | 5-shot | 10-shot | 20-shot |
| ResNet-50 | 83.62 | 88.65±1.22 | 89.21±2.19 | 91.32±2.36 | 69.70 | 71.12±11.00 | 77.53±2.85 | 78.31±2.56 | 63.65 | 75.81±13.41 | 92.38±1.22 | 94.28±1.20 |
| CNNDetection | 67.62 | 85.12±4.30 | 90.31±2.20 | 90.87±3.68 | 54.36 | 82.77±3.57 | 85.77±2.29 | 88.11±1.80 | 56.04 | 79.15±20.39 | 90.26±3.39 | 92.25±1.31 |
| LGrad | 86.48 | 71.38±3.89 | 74.86±1.94 | 78.63±1.86 | 65.61 | 63.48±8.88 | 70.88±3.15 | 73.04±2.69 | 74.23 | 73.25±2.89 | 81.40±2.20 | 83.46±2.15 |
| Gram-Net | 81.18 | 73.91±9.15 | 82.80±1.99 | 84.70±1.84 | 65.87 | 67.87±9.99 | 76.54±1.88 | 79.36±2.05 | 67.77 | 79.05±6.22 | 88.45±1.22 | 89.10±1.58 |
| FreqNet | 74.80 | 61.25±7.44 | 65.39±2.81 | 67.73±1.64 | 54.58 | 61.43±6.59 | 58.88±5.17 | 66.45±2.40 | 58.14 | 61.12±6.71 | 64.31±6.97 | 74.52±2.20 |
| NPR | 56.25 | 77.48±3.62 | 81.04±3.71 | 87.60±1.50 | 42.15 | 65.66±4.27 | 68.97±5.18 | 75.29±3.58 | 68.96 | 73.00±12.46 | 84.37±1.32 | 89.71±2.12 |
| SAFE | 83.27 | 84.02±3.59 | 88.34±0.84 | 89.98±1.26 | 68.61 | 59.79±8.58 | 68.32±3.34 | 71.03±3.09 | 67.73 | 57.10±13.22 | 72.11±4.28 | 77.95±3.27 |
| LASTED | 73.92 | 68.33±4.60 | 71.96±4.00 | 78.27±1.20 | 57.75 | 56.48±3.01 | 62.05±1.80 | 60.38±1.36 | 61.94 | 66.03±1.82 | 68.20±3.14 | 75.59±3.66 |
| UniFD | 87.63 | 92.59±3.51 | 96.68±1.33 | 98.09±0.20 | 86.67 | 86.28±2.54 | 90.41±1.89 | 92.92±1.27 | 62.66 | 96.13±0.58 | 97.75±0.78 | 98.85±0.50 |
| CLIPDetection | 94.19 | 90.73±2.54 | 93.60±0.96 | 94.48±1.00 | 73.01 | 85.75±2.85 | 86.97±2.39 | 88.38±1.46 | 60.81 | 82.26±7.80 | 85.39±6.26 | 94.79±1.41 |
| AIDE | 91.75 | 92.42±0.29 | 92.24±0.78 | 93.59±0.20 | 63.48 | 61.98±5.39 | 67.23±5.77 | 69.59±1.94 | 56.45 | 63.97±8.94 | 72.27±1.90 | 77.49±2.01 |
| DIRE | 90.02 | 91.14±3.95 | 91.98±2.72 | 92.84±1.93 | 71.11 | 81.11±15.74 | 86.99±5.47 | 89.51±0.66 | 81.35 | 87.61±6.30 | 91.81±1.91 | 93.14±0.93 |
| Ours | **96.95** | **97.90±0.86** | **98.50±0.28** | **98.68±0.14** | **98.00** | **97.95±1.58** | **99.23±0.17** | **99.50±0.14** | **96.35** | **97.52±0.66** | **98.68±0.24** | **99.04±0.28** |

*Table 9.* **Detailed few-shot SVM refitting results on ID/OOD domains trained on Community.** All methods are trained only on Community. We evaluate on three OOD test sets, Community, AIGIBench, and Chameleon. For each method, we select the intermediate layer with the best AUROC, and then train an SVM on the target domain using $N$ labeled samples per class, where $N \in \{0, 5, 10, 20\}$. The metric is AUROC (%); 0-shot denotes using the method's original classifier directly. For each $N$, results are averaged over 5 random draws and reported as mean±std. Best results for each dataset and each $N$ are **bolded**.

| Method | Community | | | | AIGIBench | | | | Chameleon | | | |
|---|---|---|---|---|---|---|---|---|---|---|---|---|
| | 0-shot | 5-shot | 10-shot | 20-shot | 0-shot | 5-shot | 10-shot | 20-shot | 0-shot | 5-shot | 10-shot | 20-shot |
| ResNet-50 | 80.49 | 81.62±1.98 | 83.18±4.80 | 88.57±2.58 | 61.12 | 62.61±10.37 | 73.85±4.85 | 75.18±2.56 | 61.15 | 66.29±8.77 | 77.19±10.79 | 86.68±2.17 |
| CNNDetection | 65.32 | 69.48±26.53 | 87.21±2.50 | 88.90±1.73 | 63.47 | 63.80±21.83 | 82.83±2.08 | 87.34±1.57 | 70.07 | 86.78±1.35 | 86.94±5.42 | 89.78±2.33 |
| LGrad | 86.01 | 64.83±4.50 | 67.68±2.77 | 73.35±1.10 | 66.81 | 63.05±5.03 | 68.19±2.35 | 69.96±3.58 | 59.66 | 59.81±8.80 | 66.09±8.72 | 77.52±3.17 |
| Gram-Net | 81.24 | 82.54±3.25 | 86.29±1.11 | 88.54±1.19 | 64.53 | 75.54±5.02 | 76.69±3.69 | 81.04±1.54 | 75.97 | 78.21±5.79 | 88.48±0.88 | 90.27±0.85 |
| FreqNet | **89.53** | 61.78±5.27 | 68.94±3.34 | 70.67±5.09 | 59.41 | 64.88±5.98 | 64.29±0.74 | 67.53±3.79 | 51.98 | 57.13±5.92 | 65.65±3.53 | 72.00±2.29 |
| NPR | 55.65 | 78.92±2.74 | 82.28±3.26 | 88.49±1.15 | 38.66 | 58.19±8.63 | 71.28±1.46 | 76.50±2.34 | 49.21 | 71.83±11.54 | 81.84±3.07 | 87.49±1.35 |
| SAFE | 73.71 | 81.93±4.93 | 86.35±0.82 | 88.89±1.38 | 65.22 | 58.93±9.76 | 69.05±2.33 | 71.25±5.56 | 78.77 | 66.20±16.42 | 78.72±4.38 | 85.80±2.19 |
| LASTED | 68.33 | 67.41±3.35 | 69.85±3.23 | 72.70±1.48 | 56.43 | 52.73±2.92 | 53.92±1.61 | 56.01±0.92 | 76.34 | 68.03±16.22 | 77.61±0.71 | 78.63±1.00 |
| UniFD | 71.57 | 92.59±3.51 | 96.68±1.33 | **98.09±0.20** | 83.85 | 86.28±2.54 | 90.41±1.89 | 92.92±1.27 | 76.78 | 96.13±0.58 | 97.75±0.78 | 98.85±0.50 |
| CLIPDetection | 88.92 | 82.44±3.22 | 88.00±1.73 | 90.39±2.27 | 63.07 | 74.73±4.11 | 78.86±1.63 | 81.87±1.12 | 63.06 | 68.80±23.94 | 81.27±0.26 | 82.28±0.76 |
| AIDE | 81.27 | 86.24±0.98 | 87.08±2.32 | 90.49±1.93 | 64.60 | 52.18±9.50 | 64.47±3.24 | 68.41±1.28 | 54.89 | 58.64±8.84 | 69.36±3.23 | 73.47±2.36 |
| DIRE | 87.09 | 88.55±3.03 | 90.15±1.89 | 93.07±1.19 | 80.22 | 82.52±5.62 | 86.26±2.03 | 89.21±0.72 | 89.30 | 87.84±1.62 | 91.46±0.82 | 92.47±0.77 |
| Ours | 86.99 | **96.50±0.98** | **97.78±0.60** | 97.96±1.18 | **93.84** | **97.57±1.66** | **98.19±0.53** | **99.13±0.16** | **96.37** | **98.38±0.51** | **98.36±0.42** | **98.99±0.26** |

