# OpenReview forum: "FiSeR: Fine-Grained Source Representations for Cross-Domain AI Image Detection"
_ICML.cc/2026/Conference — ICML 2026 regular_

### Official Review · Reviewer_2Q8t · 2026-03-01

**Soundness:** 2
**Presentation:** 2
**Significance:** 2
**Originality:** 3
**Overall Recommendation:** 4
**Confidence:** 3

**Summary:**

FiSeR addresses poor cross-domain generalization in AI image detection by revealing that feature representations remain separable despite performance drops. It proposes hierarchical contrastive learning with coarse natural-synthetic and fine generator-identity constraints, plus a k-NN homophily metric. This yields transferable representations, achieving SOTA results with minimal few-shot adaptation.

**Compliance With Llm Reviewing Policy:**

Affirmed.

**Final Justification:**

I thank the authors for their detailed response, which addressed most of my technical concerns.

**Key Questions For Authors:**

The experiments compare several methods. However, several closely related methods published at CVPR/ICCV 2024–2025 are not included. Could the authors provide additional comparisons or discuss why these methods were excluded?

**Limitations:**

Consider adding a subsection discussing scenarios where FiSeR may underperform (e.g., extreme post-processing combinations).

**Strengths And Weaknesses:**

Strengths
FiSeR learns robust feature representations to domain shifts through hierarchical contrastive learning, simultaneously optimizing coarse-grained separation of "natural vs. synthetic" and fine-grained structure of "generator identity." Combined with a lightweight classification head, it achieves state-of-the-art (SOTA) cross-domain AI image detection performance.


Weaknesses

1, The core assumption of the derivation in Appendix B is that when calculating the expected homophily for the $k$ neighbors of sample $i$, each neighbor is independently and uniformly drawn from the remaining $n-1$ samples. This assumption is crucial for arriving at the concise formula for $T_{\text{null}}(\mathcal{Z})$. However, in a real representation space and the actual computation of k-NN, a sample's $k$ neighbors are highly correlated---they are the closest points in the entire feature space to the anchor sample, and are necessarily proximate to each other in space. This induces transitivity or clustering of homophily among these neighbors, meaning the appearance of one in-class neighbor significantly increases the probability that its spatial neighbors (i.e., other points within the same set of $k$ neighbors) are also in-class, contrary to being independent events. Could the authors please clarify whether modeling the k-NN relationship via "independent draws" rather than "clustered/correlated draws" under the "random mixing" null hypothesis might overestimate the expected homophily $T_{\text{null}}$? If it is an overestimate, would the subsequent normalized homophily score $T_k(\mathcal{Z}) = \frac{T_k(\mathcal{Z}) - T_{\text{null}}(\mathcal{Z})}{1 - T_{\text{null}}(\mathcal{Z})}$ be potentially systematically underestimated, thereby introducing bias when comparing different methods?

2, The formula derivation and the final expression for $T_{\text{null}}(\mathcal{Z})$ show that under the random mixing assumption, $T_{\text{null}}(\mathcal{Z})$ is completely independent of the number of neighbors $k$. In practice, however, $T_k(\mathcal{Z})$ in a real feature space is usually sensitive to the value of $k$: a small $k$ reflects local tightness, while a larger $k$ reflects class purity over a broader region. The paper mentions in Section 4.3 that "Scores are averaged over $k\in\{1,5,10,15,20\}$", indicating an empirical selection of $k$ in practice. Does this theoretical derivation provide any theoretical insight or constraint for the optimal choice of $k$? For instance, is there a theoretical range of $k$ for which the correlation between $T_k(\mathcal{Z})$ and the final downstream detection performance (e.g., AUROC, as shown in Figure 4) is strongest? Alternatively, does the "random mixing" assumption underlying the derivation remain valid when $k$ is very large or very small relative to $n$? The paper does not discuss how the choice of $k$ affects the stability and discriminative power of this metric as a proxy for representation quality.
\end{enumerate}

3, Why weren't Acc and AUC used as evaluation metrics? It's recommended to add these two metrics.

4, The experimental evaluation would benefit from more comprehensive comparisons with several recent and relevant detectors in the literature. Specifically, the manuscript does not include direct performance comparisons or detailed discussions with the following representative methods:

① AIDE: A sanity check for AI-generated image detection. 2025 ICLR

② Effort: Orthogonal subspace decomposition for generalizable AI-generated image detection. 2025 ICML

③ Any-resolution ai-generated image detection by spectral learning. 2025 CVPR

④ C2p-clip: Injecting category common prompt in clip to enhance generalization in deepfake detection. 2025 AAAI

⑤ PiD: Generalized ai-generated images detection with pixelwise decomposition residuals. 2025 PMLR

⑥ Secret lies in color: Enhancing ai-generated images detection with color distribution analysis. 2025 CVPR

⑦ LOTA: Bit-planes guided ai-generated image detection. 2025 ICCV

---

> ### Author Rebuttal · Authors · 2026-03-31
>
> We thank the reviewer 2Q8t for the careful reading and positive assessment of the paper's method design, and experimental results. We understand that the main concerns are the theoretical derivation of the homophily metric in Appendix B, the choice of $k$, comparisons with recent baselines, and potential failure cases. We address these points below and will revise the paper accordingly in the camera-ready version if accepted.
>
> **We also provide additional experimental results in the supplementary PDF at the anonymous link : https://anonymous.4open.science/r/FiSeR-E58B/FiSeR_rebuttal.pdf**
>
> > Weakness 1: The core assumption of the derivation in Appendix B is that when calculating the expected homophily for the neighbors of sample , each neighbor is independently and uniformly drawn from the remaining samples.[...]
>
> Thank you for this careful analysis. We would like to clarify that Appendix B does **not** assume that the $k$ neighbors are independently drawn. Under the random-mixing null hypothesis, $N_k(i)$ is modeled as a size-$k$ subset drawn uniformly without replacement from the other $n-1$ samples, among which $n_s-1$ share the same source as the anchor. This is a hypergeometric setting, not an independent-draw one.
>
> Although the draws are not independent, the expected probability of observing a same-source sample at each draw position is still $\frac{n_s-1}{n-1}$, as given by the hypergeometric distribution. Our derivation only uses linearity of expectation, so the conclusion for $T_{\text{null}}$ in Appendix B still holds.
>
> To further verify this, we added a Monte Carlo validation in Table 4 of the supplementary PDF. Across repeated simulations with 2/3/4 sources, different source sizes, different random geometric layouts, and different $k$, the empirical mean consistently matches the theoretical $T_{\text{null}}$, and we do not observe systematic bias.
>
> > Weakness 2: The formula derivation and the final expression for show that under the random mixing assumption, is completely independent of the number of neighbors.[...]
>
> Thank you for this suggestion. $T_k(\mathcal{Z})$ is a representation-quality metric, and different $k$ values capture neighborhood structure at different scales. For this reason, we average over multiple $k$ values rather than relying on a single scale. The intention is to make the metric less sensitive to any one particular neighborhood size, similar in spirit to averaging over thresholds in mAP.
>
> Both theoretically and empirically, the choice of $k$ does not affect $T_{\text{null}}$: the null expectation remains the same. We also added an analysis of the correlation between this metric and downstream few-shot AUROC at different $k$ values. As shown in Table 3 of the supplementary PDF, this correlation is very stable across different $k$, suggesting that the metric is not highly sensitive to the specific choice of neighborhood size in practice.
>
> > Weakness 3: Why weren't Acc and AUC used as evaluation metrics? It's recommended to add these two metrics.
>
> We would first like to clarify that all tables in the paper already report AUROC, which is the standard AUC metric for binary detection. In addition, we report TPR@FPR=5\% (TPR5\%), because our focus is cross-domain detection, where low-false-positive operating points are often more meaningful in practice than accuracy at a single threshold.
>
> We do not use Accuracy as a main metric because it depends strongly on threshold choice and calibration, both of which are often unstable under domain shift. In contrast, AUROC and TPR5$\%$ better reflect practical detector performance in this setting.
>
> > Weakness 4 & Q1: The experimental evaluation would benefit from more comprehensive comparisons with several recent and relevant detectors in the literature.[...]  & The experiments compare several methods.[...]
>
> Thank you for the suggestion. The current paper already includes comparison with AIDE. Following your recommendation, we further added experiments with Effort, C2p-CLIP, Any-resolution Spectral Learning (SPAI), and LOTA. The results are reported in Table 5 of the supplementary PDF and will also be included in the final version.
>
> For PiD and Secret Lies in Color (CoD), we have not yet found usable open-source code, so we are currently unable to reproduce them fairly under the same unified protocol.
>
> > Limitations: Consider adding a subsection discussing scenarios where FiSeR may underperform (e.g., extreme post-processing combinations).
>
> Thank you for this suggestion. We added experiments with common perturbations, including JPEG compression, Gaussian blur, and cropping. As shown in Figure 1 of the supplementary PDF, FiSeR remains overall stable under these degradations.

---

> > ### Author Rebuttal · Reviewer_2Q8t · 2026-04-01
> >
> > In real-world scenarios, images are frequently disseminated through social media platforms (e.g., WeChat, Twitter, Facebook), which often apply lossy compression, resizing, or other post-processing operations. Could the authors clarify whether the proposed detection method maintains robust performance when evaluating images that have undergone such social network transmission? Additional experiments or discussion on the method's resilience to platform-induced distortions would significantly strengthen the practical relevance of this work.

---

> > > ### Author Response · Authors · 2026-04-01
> > >
> > > Thank you for your careful review and response. We have added experiments on common social-media perturbations on the two out-of-distribution test sets, Chameleon and AIGIBench, including Gaussian blur (sigma = 0.5, 0.9, 1.3, 1.7, 2.0), JPEG compression (quality = 100, 84, 68, 52, 36), and center cropping (retained fraction = 0.9, 0.75, 0.6, 0.45, 0.3). As shown in Figure 1 of the supplementary PDF, FiSeR remains overall stable under these degradations.
> > >
> > > For convenience, **we provide the link again here: https://anonymous.4open.science/r/FiSeR-E58B/FiSeR_rebuttal.pdf**
> > >
> > > In addition, to better reflect more realistic platform-induced distortions, we construct sequential degradation chains on the most challenging out-of-domain Chameleon dataset to simulate social-media transmission and post-processing. The degradation types include platform transmission simulation, filters, sticker occlusion, crop-and-resize recovery, and screenshot simulation. In particular, platform transmission is approximated through resizing, slight blurring, JPEG recompression, and mild saturation/contrast adjustment, while the remaining operations mimic common social-media editing behaviors. For each image, we randomly apply 3–5 degradations in sequence, resulting in a compound degradation process that is more realistic than single synthetic perturbations.
> > >
> > > The detailed results are as follows:
> > >
> > > | Method        | Original | Degraded |  Drop |
> > > | -| -: | -: | -: |
> > > | Ours |    96.35 |    91.10 |  5.25 |
> > > | AIDE |    56.45 |    46.59 |  9.86 |
> > > | CLIPDetection |    60.81 |    45.76 | 15.05 |
> > > | DIRE|    81.35 |    67.29 | 14.06 |
> > >
> > > Our method remains robust under this setting and significantly outperforms the existing baselines. We will include these experimental details and results in the final version.

---

### Official Review · Reviewer_MP7R · 2026-03-10

**Soundness:** 3
**Presentation:** 3
**Significance:** 3
**Originality:** 2
**Overall Recommendation:** 3
**Confidence:** 4

**Summary:**

This paper reveals the root cause of performance degradation in cross-domain detection. It proposes a hierarchical contrastive learning strategy, a lightweight classification head, and a new k-NN based metric. The perspective is highly novel, and the empirical results are excellent.

**Compliance With Llm Reviewing Policy:**

Affirmed.

**Key Questions For Authors:**

1. Regarding the baseline results presented in the paper (e.g., in Table 2), were these results directly cited from the  publications, or were they reproduced by the authors under their unified experimental setup?
2. In Figure 5, the "natural" images, which were not subjected to fine-grained contrastive learning, are observed to form distinct clusters. These clusters appear to align with their respective data sources. Is this impressive separability primarily attributable to the powerful, inherent capabilities of the DINOv3 backbone? Additionally, could the proposed hierarchical contrastive learning framework be readily applied to other foundation models, such as CLIP-based detectors?
3. I noticed that the results for the pre-trained DINOv3 backbone (before fine-tuning) are not reported in Tables 2, 3, and 4. Similarly, a UMAP visualization for the pre-trained features (akin to Figure 5) is also absent. Could you clarify the reasoning behind this omission?
4. How robust is the proposed feature extractor to common image degradations, such as JPEG compression, Gaussian blur, or cropping? Specifically, would these transformations significantly impact the stability and discriminability of the learned feature representations?

**Limitations:**

The asymmetry of Formula 7 overlooks the fact that real images also exhibit diverse fine-grained variations, which may result in misjudgments.

**Strengths And Weaknesses:**

strength:
1. Novel and Inspiring Perspective：The paper reveals the root cause of cross-domain performance degradation in existing detectors (i.e., the overfitting of the classification head rather than the collapse of features). This highly inspiring insight breaks the conventional thinking and points out a promising new direction for future AI image detection research.
2. Innovative Methodology and Metric : The proposed hierarchical contrastive learning elegantly moves beyond simple binary classification (real vs. fake) by utilizing generator identities as fine-grained labels. By decoupling the classification head from the feature extractor, the model achieves substantial performance gains. Furthermore, the innovative introduction of the k-NN graph homophily score provides a solid and objective evaluation tool for future representation learning research.
3. Rigorous and Convincing Experiments :The empirical validation is thorough and compelling. The authors evaluate their method across multiple challenging benchmarks and compare it against a wide range of state-of-the-art baselines, firmly supporting their claims.
weakness:
1. Formula 7 only conducts fine-grained comparisons on synthetic images. In reality, real images also exhibit substantial fine-grained variations—such as those originating from different camera brands or smartphones. Overlooking this may lead to misjudgments of real images.
2. robustness evaluation against common image corruptions (or perturbations) is lacking.
3. The paper claims that only a few samples are needed for improved few-shot performance. However, in practice, the specific source of an image is often unknown, making zero-shot the more realistic scenario.

---

> ### Author Rebuttal · Authors · 2026-03-31
>
> We thank the reviewer MP7R for the strong recognition of the paper’s main claim, method design, and experimental quality. We understand that the main concerns are: whether using fine-grained supervision only on the synthetic side in Formula 7 overlooks fine-grained variations in real images, and the role of pre-trained DINOv3 and robustness under common image degradations. We address these points below and will revise the paper accordingly in the camera-ready version if accepted.
>
> **We also provide additional experimental results in the supplementary PDF at the anonymous link : https://anonymous.4open.science/r/FiSeR-E58B/FiSeR_rebuttal.pdf**
>
> > Weakness 1 & Limitations: Formula 7 only conducts fine-grained comparisons on synthetic images.[...] & The asymmetry of Formula 7 [...]
>
> Thank you for this suggestion. We also considered this issue during method design. We currently do not apply fine-grained supervision to natural images mainly because existing datasets usually do not provide fine-grained source labels for real images, such as camera or device identity. Therefore, Formula 7 is only applied to synthetic samples.
>
> At the same time, the coarse-grained loss is still applied to all natural and synthetic samples. So this design does not assume that natural images have no fine-grained structure. Rather, in the absence of natural-side source labels, it focuses on learning stable shared features on the real side. We discuss this point in lines 148--151 of the paper: due to the imaging process, real images tend to show more stable statistical patterns in low-level feature space, and we cite prior work to support this.
>
> More importantly, our results do not suggest that this asymmetric design collapses real images into a single cluster. On the contrary, as shown in Figure 2 of the supplementary PDF, compared with pre-trained DINOv3, FiSeR makes natural-image clusters from different sources more compact.
>
> If future datasets provide fine-grained source labels for natural images, we would be very interested in extending the framework to include natural-side fine-grained supervision as well.
>
> > Weakness 2 & Q4: robustness evaluation[...] & How robust is the proposed feature extractor to common image degradations, such as JPEG compression, Gaussian blur, or cropping? [...]
>
> Thank you for this suggestion. We added experiments with common perturbations, including JPEG compression, Gaussian blur, and cropping. As shown in Figure 1 of the supplementary PDF, FiSeR remains overall stable under these degradations.
>
> > Weakness 3: The paper claims that only a few samples are needed [...]
>
> We would like to clarify that few-shot adaptation is neither a requirement of our method nor the main evaluation setting of the paper. Our main results (Table 2 and Appendix Table 5) are strict zero-shot cross-dataset evaluations: all methods are trained on a single source dataset (WildFake or Community) and directly tested on other unseen datasets.
>
> In the few-shot setting, we only use a small number of target labels to refit a lightweight classification head. As shown in Figure 3 and Appendix Tables 6 and 7, existing methods also improve clearly from zero-shot to few-shot. This further supports our main point: under domain shift, a major bottleneck is the mismatch of the classification head, rather than a complete failure of the backbone representation.
>
> > Q1: Regarding the baseline results presented in the paper [...]
>
> All baseline results were reproduced under a unified experimental protocol. As described in Section 4.2, all methods are trained on the same source dataset and directly evaluated on multiple OOD benchmarks. We also align preprocessing across methods by following the official settings or the AIGIBench implementation. More detailed reproduction settings are provided in Appendix A.
>
> > Q2 & Q3: In Figure 5, the "natural" images, which were not subjected to fine-grained contrastive learning, [...] & I noticed that the results for the pre-trained DINOv3 backbone [...]
>
> Thank you for these questions. The clustering structure of natural images in Figure 5 is indeed partly related to the representation quality of the DINOv3 backbone. Tables 2, 3, and 4 mainly focus on task-trained representations, so we did not separately include raw pre-trained DINOv3 there. For the same reason, Figure 5 was intended to highlight the representation learned after FiSeR training.
>
> To address this point, we have added both quantitative results and UMAP visualizations for pre-trained DINOv3 (Table 2 and Figure 2 in the supplementary PDF). The results show that pre-trained DINOv3 already produces some natural-side clustering, while FiSeR further makes clusters from different sources more compact and brings clear performance gains.
>
> In addition, we applied FiSeR to CLIP-ViT-L/14, DFNB-CLIP-ViT-L/14, and DINOv2-Large, and observed consistent improvements in all cases. This suggests that the framework transfers well across different backbones.

---

> > ### Author Rebuttal · Reviewer_MP7R · 2026-04-02
> >
> > While the authors have added robustness experiments with common perturbations (e.g., JPEG compression, Gaussian blur, and cropping), I remain concerned that these controlled degradations may not fully reflect real-world image distribution shifts.
> > In practical scenarios, especially for images disseminated through social media platforms (e.g., WeChat, Twitter, Facebook), images typically undergo compound transformations, including multiple rounds of compression, resizing, format conversion, and platform-specific post-processing pipelines. These transformations often introduce non-uniform and content-dependent artifacts, which are fundamentally different from the synthetic perturbations considered in the current experiments.
> >
> > Therefore, it remains unclear whether the proposed method can maintain stable performance under such realistic, platform-induced distortions. Without evaluation on more representative real-world transmission settings (or at least a more faithful simulation of such pipelines), the practical robustness and deployment readiness of the method are still insufficiently validated.

---

> > > ### Author Response · Authors · 2026-04-02
> > >
> > > Thank you for the valuable suggestion. We agree that isolated, controlled perturbations such as JPEG compression, Gaussian blur, and cropping are insufficient to fully capture the more complex degradations encountered in real-world social media transmission. To address this concern, we further added a compound degradation experiment that better reflects practical scenarios.
> > >
> > > Specifically, we construct sequential degradation chains on the most challenging out-of-domain Chameleon dataset to simulate social-media transmission and post-processing. The degradation types include platform transmission simulation, filters, sticker occlusion, crop-and-resize recovery, and screenshot simulation. In particular, platform transmission is approximated through resizing, slight blurring, JPEG recompression, and mild saturation/contrast adjustment, while the remaining operations mimic common social-media editing behaviors. For each image, we randomly apply 3–5 degradations in sequence, resulting in a compound degradation process that is more realistic than single synthetic perturbations.
> > >
> > > The detailed results are as follows:
> > >
> > > | Method        | Original | Degraded |  Drop |
> > > | -| -: | -: | -: |
> > > | Ours |    96.35 |    91.10 |  5.25 |
> > > | AIDE |    56.45 |    46.59 |  9.86 |
> > > | CLIPDetection |    60.81 |    45.76 | 15.05 |
> > > | DIRE|    81.35 |    67.29 | 14.06 |
> > >
> > > Our method remains robust under this setting and significantly outperforms the existing baselines. We will include these experimental details and results in the final version.

---

### Official Review · Reviewer_Avs5 · 2026-03-12

**Soundness:** 3
**Presentation:** 3
**Significance:** 2
**Originality:** 2
**Overall Recommendation:** 4
**Confidence:** 4

**Summary:**

In this paper, the authors analyze why synthetic image detectors break under domain shift. While the authors find that natural and synthetic features remain partly separable on unseen datasets, accuracy still drops, suggesting the classification head overfits to the training data and identify the need to provide better representations to handle with. To combat this, they propose FiSeR, which learns more transferable representations via hierarchical supervised contrastive learning objective.  At test time, they freeze the backbone and use a lightweight plug-and-play head, optionally with few-shot target labels. They also introduce a k-NN graph homophily score to measure separability and show it correlates strongly with few-shot AUROC.  The authors perform extensive experiments on several popular benchmarks and report performance improvements.

**Compliance With Llm Reviewing Policy:**

Affirmed.

**Final Justification:**

Given the authors convincing rebuttal, I am increasing my score to weak accept.

**Key Questions For Authors:**

1. Could the authors clarify the primary novelty of this work? The approach appears closely related to supervised contrastive learning, with additional objectives to learn an encoder that generalizes to synthetic shift detection when combined with a downstream classifier. A clearer distinction would be helpful.
2. Could the authors provide more details about the generators used during training? While the loss formulation is intuitive, it seems that the method may require generative models to be trained and available for datasets such as WildCam before deployment. If so, this could introduce scalability challenges when extending to new datasets. Alternatively, does the method use generative models or is it just a representative term? How are negative examples constructed in practice?
3. Have the authors evaluated the approach using larger-scale vision backbones? It would be helpful to understand whether domain shift generalization issues persist at scale, or whether performance gaps narrow with increased model capacity.
4. How does FiSeR perform when generator identity labels s(x) are noisy or partially incorrect? An analysis of robustness to label noise would strengthen the work.
5. How sensitive are the results to key hyperparameters such as \lambda and \tau? Additional ablation studies examining this sensitivity would be valuable for a more complete evaluation.

**Limitations:**

yes

**Strengths And Weaknesses:**

Strengths:
The paper is clearly written and easy to follow. The objective functions are intuitive, and the authors conduct extensive empirical evaluations and relevant ablation studies. The experimental section is thorough and helps build confidence in the results.

Weaknesses:
Although the experiments are comprehensive, the method currently appears to be closely related to supervised contrastive learning. It may strengthen the paper if the authors could more clearly articulate the core novelty and frame the overall story in a way that highlights the key conceptual contribution.

---

> ### Author Rebuttal · Authors · 2026-03-31
>
> We thank the reviewer Avs5 for the positive assessment of the paper's writing, experimental coverage, and result credibility. We understand that the main concerns are the distinction from supervised contrastive learning and the clarification of practical assumptions and additional analysis. We address these points below and will revise the paper accordingly in the camera-ready version.
>
> **We provide additional experimental results in the supplementary PDF at the link : https://anonymous.4open.science/r/FiSeR-E58B/FiSeR_rebuttal.pdf**
>
> > **Weakness & Q1:** Although the experiments are comprehensive, the method currently appears to be closely related to supervised contrastive learning.[...]  & Could the authors clarify the primary novelty of this work? [...]
>
> Thank you for pointing this out. The key contribution of FiSeR has three closely connected parts.
>
> First, we reveal a key observation in cross-domain generative image detection: a drop in performance does not always mean that the intermediate representation has collapsed. In many cases, the feature space still keeps useful separable structure, while the classification head becomes the main bottleneck. This is shown by the UMAP analysis in lines 42-51 of the Introduction and by the few-shot head refitting results in Table 3 and Figure 3.
>
> Second, based on this observation, FiSeR is not designed to learn only a stronger closed-set classifier. Its goal is to learn a feature representation that is more stable and transferable under domain shift. FiSeR explicitly uses the hierarchical structure that already exists in AI image detection: images have a natural/synthetic coarse label, and synthetic images also contain fine-grained source differences from different generators. Standard supervised contrastive learning uses only flat labels, while FiSeR uses hierarchical contrastive learning to model both levels. This makes the supervision better matched to the data structure itself. As a result, FiSeR learns representations that generalize better to unseen generators and unseen domains. Consistent with this, Table 2 shows that FiSeR achieves SOTA results on all five datasets, and Figures 5 and 6 show that FiSeR learns clearer source-level structure and more transferable representations.
>
> Third, beyond the training objective itself, we also introduce k-NN graph homophily as a low-variance proxy for representation separability and few-shot adaptability.
>
> > **Q2:** Could the authors provide more details about the generators used during training? [...]
>
> FiSeR does not require training, accessing, or running any generative model. The generator identity / source label (e.g., DALL-E 3, Stable Diffusion 1.5) is simply a discrete label provided by the public training dataset. In practice, the fine-grained loss is applied only to synthetic samples: synthetic samples from the same source form positive pairs, while synthetic samples from different sources form negative pairs.
>
> At deployment time, FiSeR only uses the trained image encoder and a lightweight classification head. Regarding scalability, Tables 2 and 5 already provide detailed cross-dataset results, showing that FiSeR can generalize to other unseen datasets even when trained on only one dataset.
>
> > **Q3:** Have the authors evaluated the approach using larger-scale vision backbones? [...]
>
> Thank you for this suggestion. We added experiments with a larger DINOv3 ViT-H/16+ backbone under the same training setting. As shown in Table 2 of the supplementary PDF, without training, ViT-H/16+ improves over ViT-L/16 on AIGIBench, but performs worse on Chameleon. After fine-tuning, ViT-H/16+ is still slightly worse than ViT-L/16 on both datasets.
>
> These results suggest that cross-domain generalization depends more on how well the training objective encourages transferable representations, rather than on backbone size alone.
>
> > **Q4:** How does FiSeR perform when generator identity labels $s(x)$ are noisy or partially incorrect? [...]
>
> In FiSeR, noise in source labels mainly affects the fine-grained objective, which may weaken source-level structure learning but does not directly disrupt the main natural/synthetic separation. This is consistent with our ablation: removing the fine-grained loss reduces OOD performance, but the model remains effective.
>
> Since source labels usually come from dataset metadata, the actual noise level is expected to be limited. Due to rebuttal time constraints, we have not yet completed a full noise study, but we will include a label-noise robustness experiment in the final version.
>
> > **Q5:** How sensitive are the results to key hyperparameters such as $\lambda$ and $\tau$? [...]
>
> Due to training cost, we tested two representative settings (e.g., $\tau = 0.05$ and $\lambda = 2$). Table 1 in the supplementary PDF shows similar performance to the default setting across multiple OOD benchmarks, with slight gains in some cases, suggesting that FiSeR is not highly sensitive to these hyperparameters.

---

> > ### Author Rebuttal · Reviewer_Avs5 · 2026-04-03
> >
> > I thank the authors for their responses; a lot of my questions have been addressed.  I will update my rating.
> >
> > Nevertheless, I am not happy that the authors added an external link. This seems unfair to other authors who respect the 5000-character limit. I am aware that the policy allows non-anonymous links, and hence this is not a violation of the policy as per my reading, but could have been avoided.

---

> > > ### Author Response · Authors · 2026-04-04
> > >
> > > Thank you for your careful review of our response. We are glad that most of your questions have been addressed, and we will incorporate the relevant revisions and improvements into the final version. We also sincerely appreciate your comment regarding the use of the external link. We will be more careful about this in future submissions.

---

### Decision · Program_Chairs · 2026-04-30

**Decision:**

Accept (regular)

**Comment:**

The paper proposes a feature learning method for AI-generated image detection using fine-grained generator identities. It received mixed reviews, including two weak accepts and one weak reject. The rebuttal successfully addressed concerns regarding experimental protocols, provided additional results with larger backbones and more complex image perturbations, and clarified several other points.

One remaining concern is robustness in real-world scenarios, where images often undergo multiple stages of compression and distortion.

The AC has read the reviews and the rebuttal and considers the paper to be a timely contribution, given the increasing prevalence of AI-generated images on the web. Although the proposed method is relatively simple, it demonstrates consistent and notable performance improvements. The concern regarding robustness to compounded transformations is partially addressed in the rebuttal and is not considered a critical weakness. Therefore, the AC recommends acceptance.